# Secondary extended mannan side chains and attachment of the arabinan in mycobacterial lipoarabinomannan

Shiva K. Angala[1], Wei Li[1], Claudia M. Boot 🄳 [2], Mary Jackson 🄳 [1,3✉] & Michael R. McNeil[1,3]

Mycobacterial lipoarabinomannan (LAM) is in an essential cell envelope lipopolysaccharide anchored both to the plasma and outer membranes. To understand critical biological questions such as the biosynthesis, spatial organization of LAM within the cell envelope, structural remodeling during growth, and display or lack of display of LAM-based antigenicity requires a basic understanding of the primary structure of the mannan, arabinan and how they are attached to each other. Herein, using enzymatic digestions and high-resolution mass spectrometry, we show that the arabinan component of LAM is attached at the non-reducing end of the mannan rather than to internal regions. Further, we show the presence of secondary extended mannan side chains attached to the internal mannan region. Such findings lead to a significant revision of the structure of LAM and lead to guidance of biosynthetic studies and to hypotheses of the role of LAM both in the periplasm and outside the cell as a fundamental part of the dynamic mycobacterial cell envelope.

[1] Mycobacteria Research Laboratories, Department of Microbiology, Immunology and Pathology, Colorado State University, Fort Collins, CO 80523, USA. [2] Central Instrument Facility, Department of Chemistry, Colorado State University, Fort Collins, CO 80523, USA. [3] These authors jointly supervised this work: Mary Jackson, Michael R. McNeil. ✉email: Mary.Jackson@colostate.edu

Lipoarabinomannan (LAM) and its non-arabinosylated precursor, lipomannan (LM), are found abundantly in the cell envelope of mycobacteria in either the plasma membrane or outer membrane[1] (also referred to as "mycomembrane"). These lipoglycans are essential constituents of all mycobacteria, playing various important roles in cell envelope integrity[2], while at the same time modulating key aspects of the host innate and adaptive immune responses to mycobacterial infections[3–5]. LAM is generally considered to contain four structural domains: the phosphatidylinositol anchor, to which is attached the mannan backbone, followed by the arabinan, and at the non-reducing end of the arabinan various termini commonly known as caps (Fig. 1). LAM has been studied extensively, both from a fundamental and translational standpoint, for its immunological properties[1] and, more recently, as a diagnostic marker of tuberculosis in urine and serum[6–8]. Enzymes involved in the biosynthesis of LAM are further the object of intense research in the context of antimycobacterial drug development[1]. Most recently, changes in the mycobacterial envelope, of which LAM is a fundamental component, have been studied from a gene-regulation perspective within the context of the envelope being a dynamic component that undergoes changes during the bacterial life cycle and in response to its environment[9].

The structural analysis of LAM has a long history of exceptional investigations. The mannosylated phosphatidylinositol structural work dates back to the 1960s with elegant analyses by Clinton Ballou's group[10,11] elucidating the structure of the basic phosphatidylinositol mannosides, PIMs. Subsequently, arabinomannans of *Mycobacterium* spp. were studied, and although the phosphatidylinositol anchor was not recognized, the basic idea of an α-1,6 mannan backbone to which an arabinan was attached emerged from work by Misaki, Azuma, and Yamamura[12]. Shirley Hunter and Patrick Brennan then recognized that these

arabinomannans were attached to phosphatidylinositol[13,14], and more precisely to the 6 position of the mannosyl residue that itself attached at *O*-6 of inositol of $PIM_2$[15]. The location of the acyl functions was shown to be the same on LAM and on phosphatidylinositol dimannosides by Germain Puzo's group[16]. The general structure of a six-linked mannan backbone substituted at *O*-2 with terminal α-Man*p* residues was also recognized at that time[15,17,18]. Structural studies by Chatterjee and others focused mostly on the arabinan[19,20] and most especially on the non-reducing terminal "caps" of the arabinan[17,21–23]. More recently, genetic approaches to LAM biosynthesis have yielded more structural information about the mannan backbone, including the fact that the first 5–7 reducing end Man*p* residues coming out from inositol are unbranched[24].

Despite all this research, details about the attachment of the arabinan to the mannan have remained elusive. Recently, we obtained evidence that generally, a single arabinan chain is attached to the phosphatidylinositol mannosyl core of LAM by analysis of the mono-arabinosyl LM (Msm-Ara-LM) produced by an *M. smegmatis* strain in which the gene for the α-1,5 arabinosyltransferase, known as EmbC (*MSMEG_6387*), was knocked out[24]. However, details of the attachment of the arabinosyl residue to LM were not determined. Earlier, the reducing end of the arabinan of LAM was reported by one of us[17] to be attached to *O*-2 of one of the six-linked mannosyl residues based on partial hydrolysis experiments. However, recent biosynthetic studies[25] revealed that an arabinosyl residue was added by an enzyme extract of *M. smegmatis* to the six-position of the non-reducing end mannosyl residue present in the octyl glycoside of α-Man-(1→6)-Man, suggesting the possibility that the arabinan is attached to *O*-6 of a mannosyl residue in LAM. This biosynthetic observation reflected an earlier suggestion by Hunter and Brennan, where these authors suggested attachment of Ara*f* to

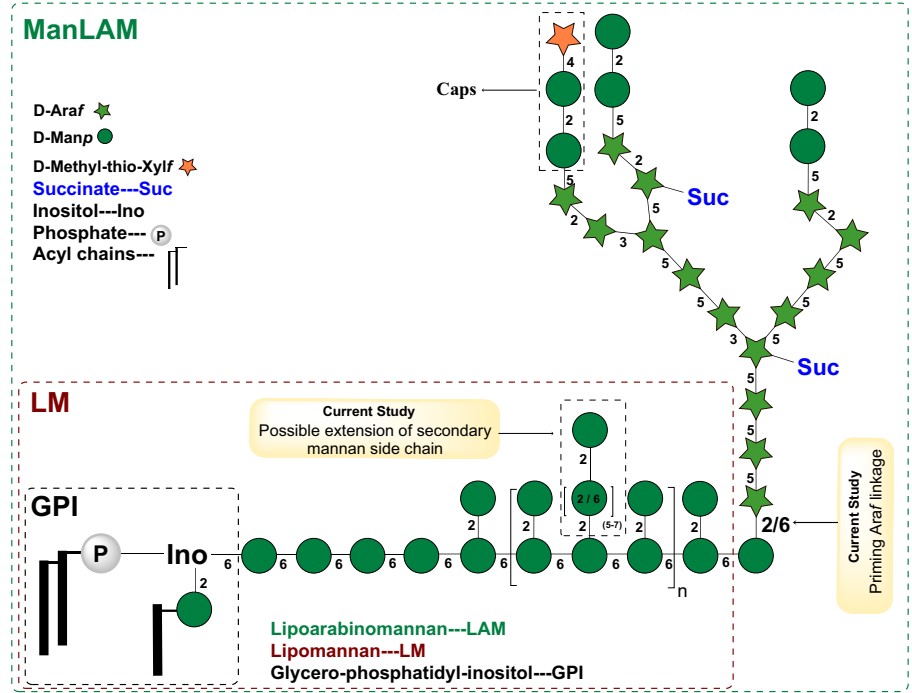

**Fig. 1 Schematic representation of mannose-capped lipoarabinomannan from *M. tuberculosis*.** Mannose-capped lipoarabinomannan (ManLAM) consists of four structural domains: (i) a phosphatidyl-myo-inositol (GPI) anchor, (ii) to which is attached mannan backbone composed of linear α-(1→6) and α-(1→2) branch points to form lipomannan (LM), (iii) which is in turn attached by the arabinan domain (composed of t-Ara*f*, 2-linked Ara*f*, 5-linked Ara*f*, and 3,5-linked Ara*f* residues) at the non-reducing end to form lipoarabinomannan (LAM), and finally (iv) substitution of capping residues (mono-, di-, or trimannosides further substituted or not with a 5'methyl-thio-Xyl*f* residue [MTX]) at the non-reducing end of the arabinan domain. Succinates are found on the internal 3,5-linked Ara*f*, and terminal 2-linked Ara*f* residues. Symbolic nomenclature for glycans was used.

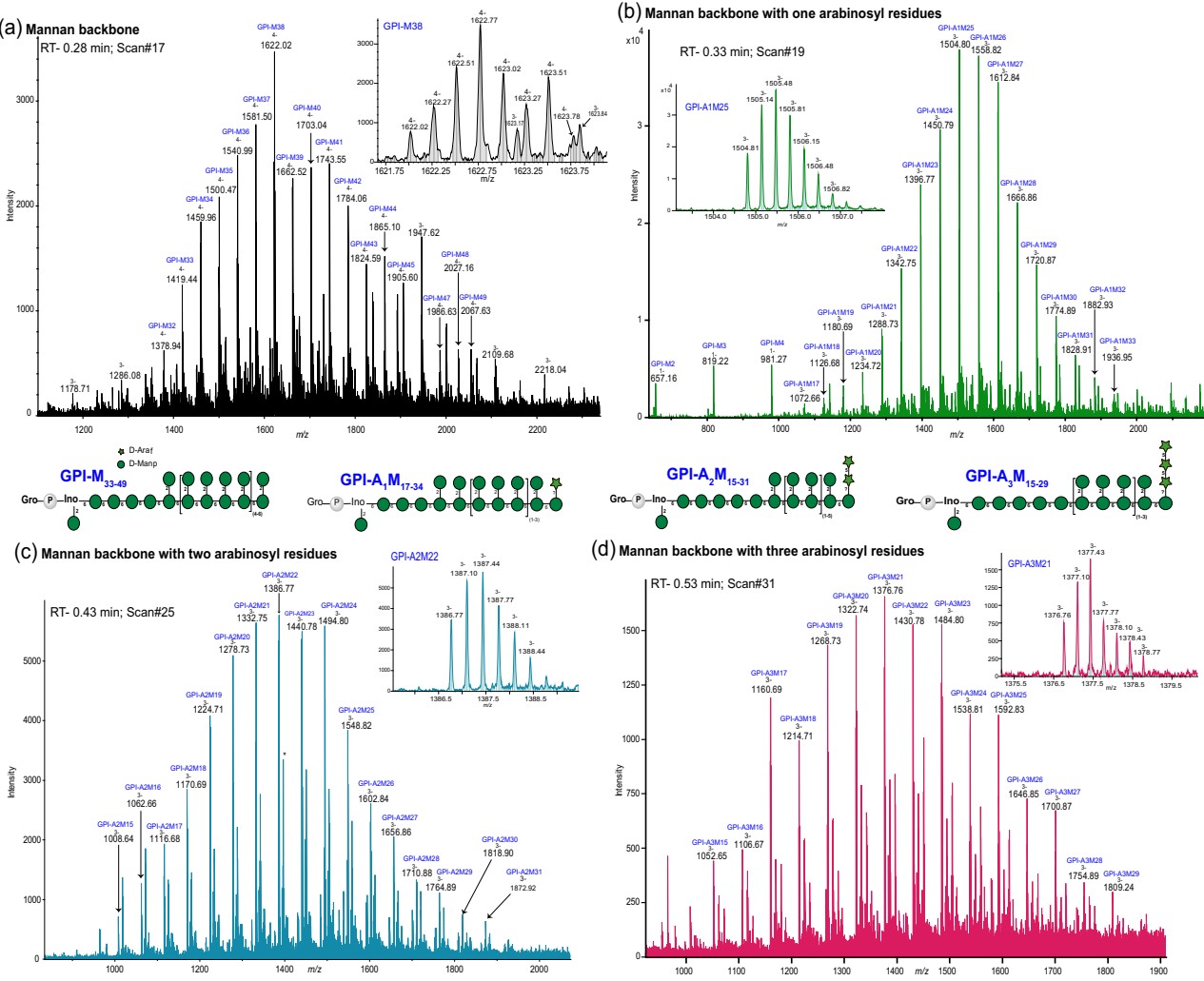

**Fig. 2 High-resolution electrospray LC–MS of deacylated "Msm-Ara-LM" produced by *M. smegmatis* Δ*MSMEG_6387* that lacks the EmbC α-1,5 arabinosyltransferase. a** Scan 17 is enriched in non-arabinosylated LM. **b** Scan 19 is enriched in Ara1-LM. **c** Scan 25 is enriched in Ara2-LM, and **d** Scan 31 is enriched in Ara3-LM. The insets in the middle panel represent the schematic structures of the mannan backbone with a varying number of arabinosyl residues. The ion intensities (*y* axis) show that, as previously reported[24], the Ara1-LM species are quantitatively dominant. Representative blowups of individual ion clusters are shown in the insets. The *m/z* labels shown are for the $^{12}$C isotope rather than the most intense component of the ion cluster.

position 6 of a mannosyl residue[13] (although their data did not rule out attachment of the Ara*f* to the 2-position of a mannosyl residue).

To resolve this issue, we here isolate an oligosaccharide where an arabinosyl residue was attached to mannosyl backbone residues, determine its structure, and, finally, elucidate the structure of the region of the mannan core around it. In this process, unexpected additional structural information concerning the mannan backbone is revealed in that extended mannan side chains are present on the mannan backbone.

## Results

**Mass spectral analysis of arabinosylated LM.** Initially, we started with Msm-Ara-LM prepared from *M. smegmatis* Δ*MSMEG_6387*, which lacks the EmbC α-1,5 arabinosyltransferase[24]. Msm-Ara-LM was deacylated and analyzed by high-resolution electrospray liquid chromatography–mass spectrometry (LC–MS) (Fig. 2, Supplementary Fig. 1, and Supplementary Table 1), which provided more details regarding the composition of this material than the matrix-assisted laser desorption ionization–time-of-flight (MALDI–TOF) MS previously revealed[24]. The major components were mono-arabinosylated LM with compositions of

Ara1-Man17–33-Inos-P-Gro with the most abundant components being Ara1-Man22–29-Inos-P-Gro (Fig. 2b). Lower-intensity ions corresponding to Ara2-Man15–31 (Fig. 2c) and smaller amounts of Ara3-Man15–30 (Fig. 2d) were also found at slightly different retention times than the main mono-arabinosylated components. Finally, non-arabinosylated LM was found in small amounts from Man32–49-Inos-P-Gro (Fig. 2a). The reason for the large size of these non-arabinosylated LM-like molecules compared with *M. smegmatis* LM that contains 21–34 Man residues[26], is unknown. Although "Msm-Ara-LM" is much more complex than originally reported[24], it is still a useful material for enzymatic digestion to isolate the Ara-Man-linkage region.

**Isolation of a small arabinosyl-containing oligosaccharide linked to the mannan backbone from Msm-Ara-LM.** To isolate an oligosaccharide containing the arabinosyl residue attached to mannosyl residues, Msm-Ara-LM was treated with a mixture of enzymes containing both commercial Jack bean α-mannosidase and the endo-α-1,6-mannanase originally characterized by Ballou and coworkers[27], which had recently been recombinantly produced and further characterized by our research group[28]. The enzyme digest was then chromatographed on a P-2 column, and

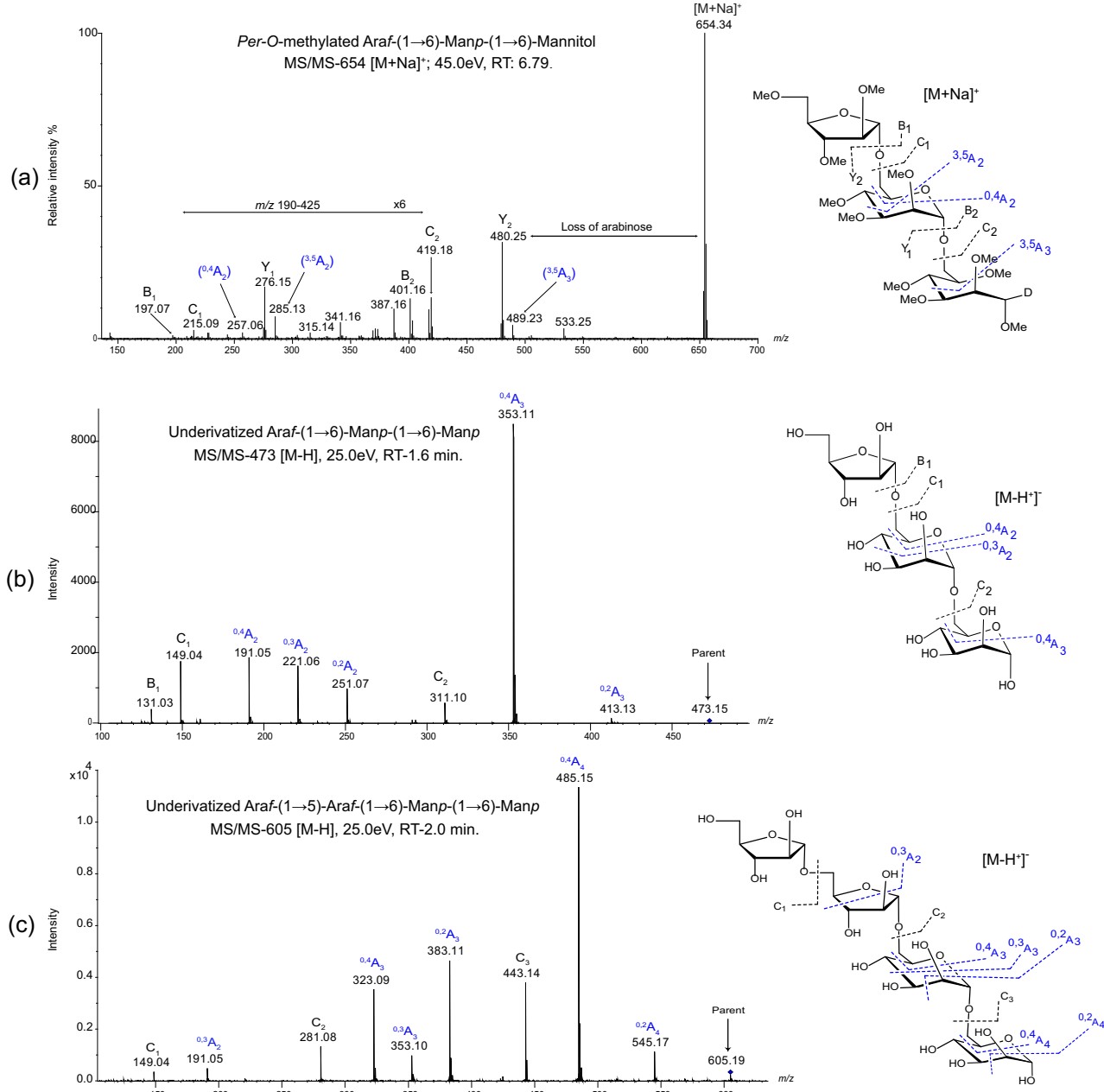

**Fig. 3 LC–MS/MS analysis of small arabinosyl-containing oligosaccharides. a** MS/MS spectrum of the [M + Na]⁺ ion at $m/z$ 654.34 of reduced and methylated Ara$f$-(1→6)-Man$p$-(1→6)-mannitol isolated from Msm-Ara-LM produced by *M. smegmatis ΔMSMEG_6387*. (**b**) MS/MS spectrum of the [M–H]⁻ ion at $m/z$ 473.15 of underivatized Ara$f$-(1→6)-Man$p$-(1→6)-Man$p$ and **c** MS/MS spectrum of the [M–H]⁻ ion at $m/z$ 605.19 of underivatized Ara$f$-(1→5)-Ara$f$-(1→6)-Man$p$-(1→6)-Man$p$. The oligosaccharides shown in (**b, c**) resulted from *Cellulomonas* endo-arabinanase-digested *M. tuberculosis* LAM after digestion with α-mannosidase and endo-1,6-α-mannanase. The sequences are shown by the B and C ion series illustrated, and the linkages (O-6 to mannosyl residues and O-5 to the internal arabinosyl residue) are shown by the A ion series. (The cleavages illustrating the B ion cleavage also require the loss of a proton; the cleavages illustrating the C ion cleavage require the addition of a proton).

aliquots of each fraction acetylated and analyzed by LC–MS (Supplementary Fig. 2a). The arabinosyl-containing product was found as a trisaccharide of Ara-Man-Man at $m/z$ 912.29 as the NH₄⁺ adduct of the acetylated trisaccharide (Supplementary Fig. 2b). Co-eluting with Ara-Man-Man was the disaccharide Man-Man, which evidently was not fully digested to monosaccharides by the α-mannosidase. For further purification and structural analysis, the P4 fractions containing Ara-Man-Man were reduced with NaBD₄, methylated, and identified by LC–MS as the Na⁺ adduct ion at $m/z$ 654.34 (Supplementary Fig. 3a, b). This ion was fragmented by collision-induced fragmentation and

yielded the spectrum shown in Fig. 3a. Using the nomenclature of Domon and Costello[29], the B and C ions clearly show a linear trisaccharide with the Ara$f$ at the non-reducing end. The assignment of the fragment ions detected in the MS/MS spectrum (positive mode) is listed in Supplementary Table 2. A ions were also produced that are consistent with both the internal mannosyl and the reducing mannitol residue being linked at O-6.

To confirm the linkage, the per-O-methylated Ara-Man-mannitol was purified by LC free of the Man-mannitol. The purified product was hydrolyzed, reduced, and acetylated along with standards of α-Man-(1→6)-mannitol, α-Man-(1→4)-

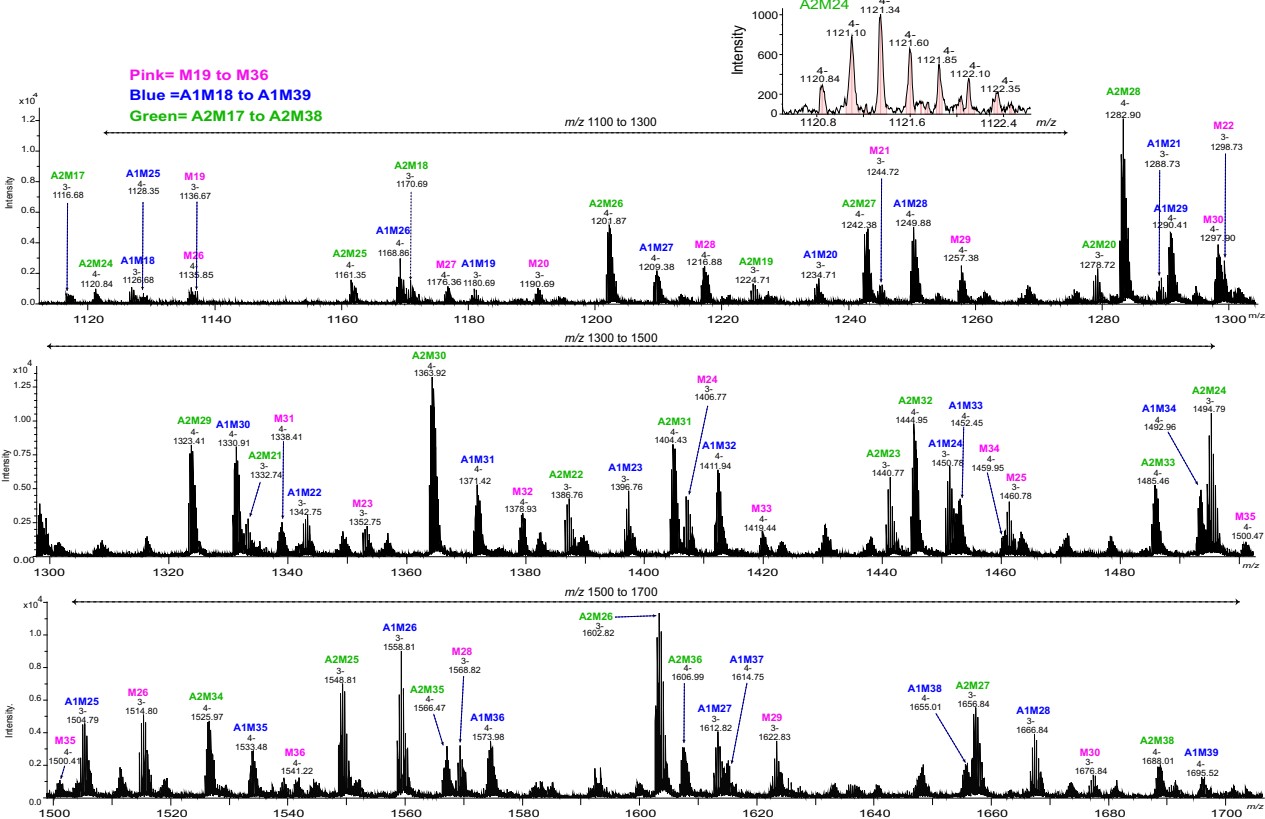

**Fig. 4 High-resolution electrospray LC–MS of the Ara$_{0-2}$-LM formed from *M. tuberculosis* LAM by *Cellulomonas* endo-arabinanase digestion.** In this case, the molecules with various degrees of arabinosylation did not readily separate, and an average spectrum is presented. The degrees of arabinosylation and ionization (both triple- and quadruple-charged) are marked in the figure and color-coded for degree of arabinosylation. Ions produced by Ara$_3$-LM are present in low amounts and not identified.

mannitol, and α-Man-(1→2)-mannitol, and β-Glc-(1→6)-Glucitol. The MS spectra clearly showed 1,4-di-*O*-acetyl-2,3,5-tri-*O*-methyl-arabinitol, 6-*O*-acetyl-1,2,3,4,5-penta-*O*-methyl-mannitol, and 1,5,6-tri-*O*-acetyl-2,3,4-tri-*O*-methyl mannitol as illustrated in Supplementary Fig. 4. Importantly, no 2-*O*-acetyl-1,3,4,5,6-penta-*O*-methyl-mannitol or 1,2,5-tri-*O*-acetyl-3,4,6-tri-*O*-methyl mannitol were detected. Thus, the trisaccharide Ara*f*-(1→6)-Man*p*-(1→6)-Man was unequivocally identified, thereby demonstrating that our earlier report[17] was in error, and in fact the arabinan is attached to the mannan backbone at the *O*-6 of a mannosyl residue. Unfortunately, an attempt to determine the anomeric configuration of the trisaccharide by nuclear magnetic resonance (NMR) was unsuccessful due to insufficient amounts of material.

**Preparation of arabinan-deficient LAM from *M. tuberculosis* and isolation and characterization of small arabinosyl-containing oligosaccharides.** To determine whether a similar attachment of the arabinan to the mannan backbone was found in the major mycobacterial pathogen, *M. tuberculosis*, LAM isolated from this bacterium was deacylated and treated with *Cellulomonas* arabinanase[21] and the mannan region with its attached arabinosyl "stubs" isolated by LC. Electrospray LC–MS revealed the presence of a complex mixture of glycerolphosphoinositol mannans, which were substituted with 0–3 arabinosyl residues with the substitution of 1 and 2 arabinosyl residues most dominant (Fig. 4; Supplementary Fig. 5; Supplementary Table 3). This result was to be expected as the degree of de-arabinosylation by the *Cellulomonas* endo-arabinanase is not controllable. However, the

Ara$_1$- and Ara$_2$-substituted molecules were expected to yield the arabinosyl–mannosyl-linkage unit.

The *Cellulomonas* endo-arabinanase-digested *M. tuberculosis* LAM was then digested with a mixture of enzymes containing both α-mannosidase and the endo-α-1,6-mannanase. Both tri Ara-Man-Man and tetra Ara-Ara-Man-Man oligosaccharides were identified by LC–MS analysis (Supplementary Fig. 3c, d). LC–MS/MS on the nonderivatized oligosaccharides (Fig. 3b, c; Supplementary Table 2) showed the trisaccharide to be Ara*f*-(1→6)-Man*p*-(1→6)-Man*p* and the tetrasaccharide to be Ara*f*-(1→5)-Ara*f*-(1→6)-Man*p*-(1→6)-Man*p*.

**Determination of the attachment site of the arabinosyl residue to the mannosyl region.** We next performed experiments to determine the original substitutions (or lack of substitutions) on *O*-2 of the mannosyl residues present in the Ara*f*-(1→6)-Man*p*-(1→6)-Man trisaccharide before possible removal of such residues by the α-mannosidase.

To address this issue, Msm-Ara-LM was methylated and analyzed by LC–MS/MS. Although both B and Y ions (see Fig. 3a for their formation) were produced, they were not informative for the question of further substitution of the mannosyl residue to which the arabinosyl residue is attached. However, at low mass, low-intensity oxonium ions were found as illustrated in Fig. 5 and Supplementary Fig. 6. The intensities of low-mass oxonium ions in the MS/MS spectrum of the methylated Ara-Man$_{23}$-Inos-P-Gro triply charged (NH$_4$$^+$)$_3$ ion at *m/z* 1784.89 are presented in Table 1. These data suggest that the mannose to which the Ara is attached also has a mannosyl residue at *O*-2, given the lack of ion at *m/z* 379 (Ara-Man$_1$) and

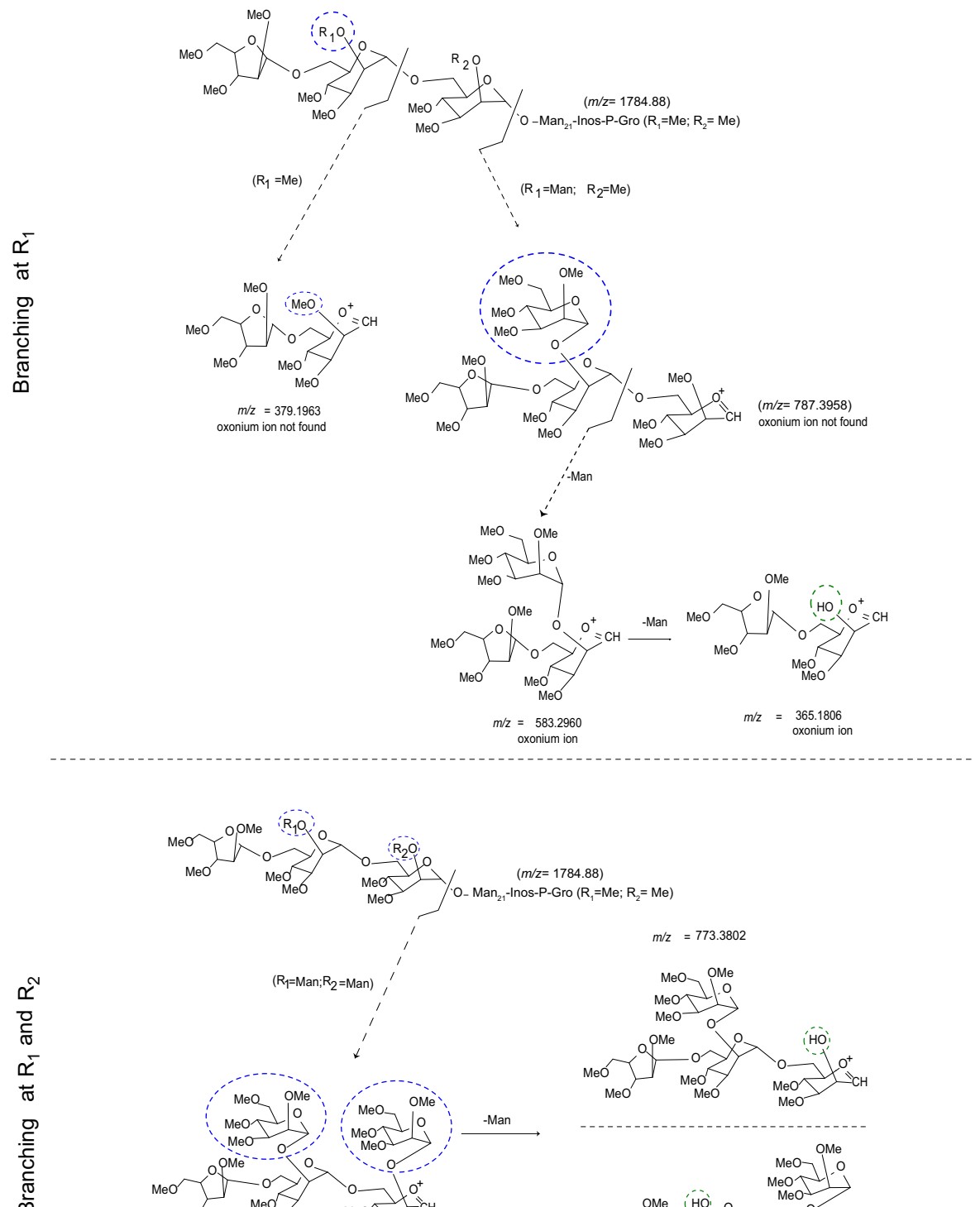

**Fig. 5 The formation and *m/z* values of possible oxonium ions formed during MS/MS of the methylated Ara$_1$-Man$_{23}$-Inos-P-Gro triply charged (NH$_4^+$)$_3$ ion at *m/z* 1784.89.** The MS/MS spectrum of *m/z* 1784.89 is described in Supplementary Fig. 6.

the presence of ions at *m/z* 583 (Ara-Man$_2$) and *m/z* 365 (Ara-Man$_1$ with an OH group due to the loss of substituent at *O*-2). A similar pair of ions are found at *m/z*'s 991 and 773. Along with the lack of an ion at 787, this finding suggests that before

α-mannosidase treatment, both mannosyl residues of the isolated trisaccharide, Ara*f*-(1→6)-Man*p*-(1→6)-Man, were substituted with α-Man*p* at *O*-2. Due to the low intensity of these ions, we cannot conclude that these mannosyl residues are

**Table 1 Intensities of oxonium ions at the non-reducing end terminal of per-*O*-methylated *M. smegmatis* Ara₁-Man₂₃-Ino-P-Gro observed in MS/MS of the triply charged (NH₄⁺)³⁺ ion at *m/z* 1784.88.**

| Composition | Theoretical *m/z* values | Intensity (counts) | Observed *m/z* values | Mass error (Da) | Mass accuracy (ppm) |
|---|---|---|---|---|---|
| *Mass of fully O-methylated (no OH group)* | | | | | |
| $Ara_1$-$Man_4$ | 991.4956 | 142 | 991.4926 | −0.0030 | −3.0257 |
| $Ara_1$-$Man_3$ | 787.3958 | 0 | nd | nd | nd |
| $Ara_1$-$Man_2$ | 583.2960 | 347 | 583.2971 | 0.0011 | 1.8858 |
| $Ara_1$-$Man_1$ | 379.1963 | 0 | nd | nd | nd |
| *Mass of fragment ions with one O-methyl group replaced by OH group* | | | | | |
| $Ara_1$-$Man_4$ | 977.4800 | 0 | nd | nd | nd |
| $Ara_1$-$Man_3$ | 773.3802 | 420 | 773.3774 | −0.0028 | −3.6205 |
| $Ara_1$-$Man_2$ | 569.2804 | 0 | nd | nd | nd |
| $Ara_1$-$Man_1$ | 365.1806 | 1720 | 365.1791 | −0.0015 | −4.1076 |

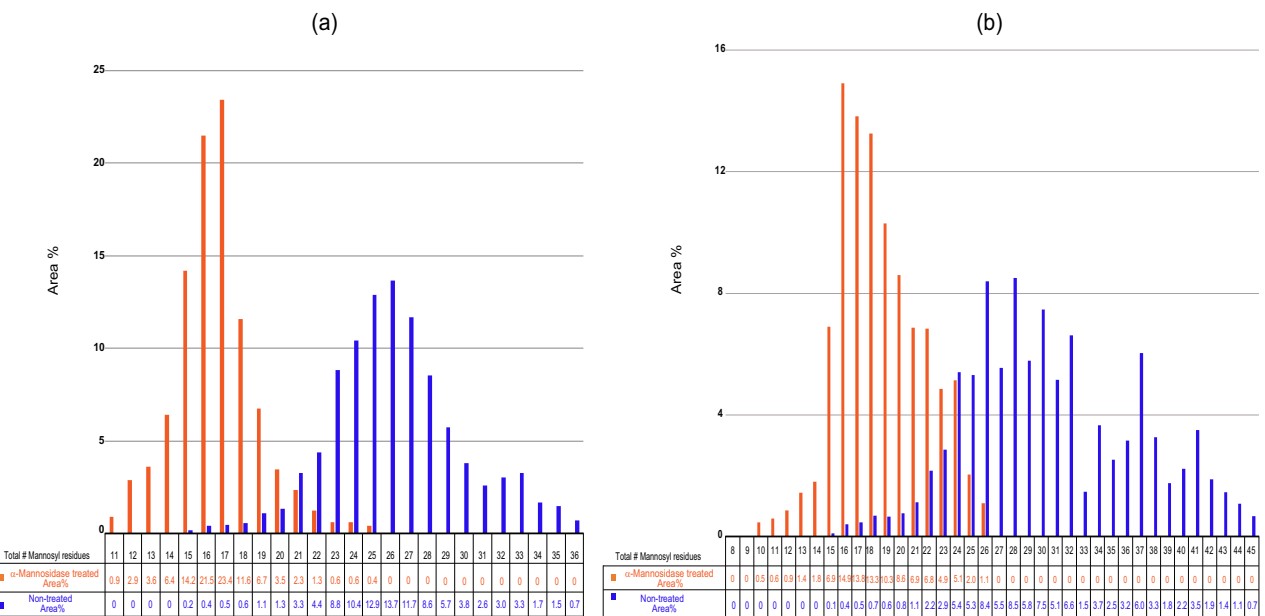

**Fig. 6 Changes in the composition of the mannan backbone after digestion with α-mannosidase. a** A comparison of the number of mannosyl residues present in Ara₁-LM produced by *M. smegmatis* ΔMSMEG_6387 before (blue) and after (orange) α-mannosidase treatment. **b** A comparison of the number of mannosyl residues present in Ara₁-LM formed from *M. tuberculosis* LAM by *Cellulomonas* endo-arabinanase digestion before (blue) and after (orange) α-mannosidase treatment. *M. smegmatis* LAM digestion was done once. The results presented for *M. tuberculosis* LAM are representative of two independent experiments. The area percentages were calculated from the MS¹ spectra as shown in Supplementary Data 1.

always substituted at *O*-2, but rather that they can be and mostly are so substituted.

**Determination of the length of the mannan backbone between the inositol and terminal Ara*f* unit by treatment of Msm-Ara-LM with Jack Bean α-mannosidase.** Initial experiments showed that α-mannosidase does not remove the mannosyl residue at *O*-2 of the inositol, and that the products released by α-mannosidase treatment of Msm-Ara-LM are linear as expected (see Supplementary Note 1; Supplementary Fig. 7). The distribution of the linear Ara-Man*x*-Inos-P-Gro molecules formed by treatment of Msm-Ara-LM with α-mannosidase is compared with the original distribution of Ara-Man*x*-Inos-P-Gro in Fig. 6a and Supplementary Data 1. The majority (78%) of enzyme-treated material contains between 14 and 19 mannosyl residues, suggesting that the α-1,6 backbone between the T-Ara*f* residue and the inositol usually contains 13–18 mannosyl residues (Supplementary Fig. 8a, c). This analysis cannot determine which α-mannosidase products came from which original Msm-Ara-LM molecules.

A similar analysis was performed on the *Cellulomonas* endo-arabinanase-digested *M. tuberculosis* LAM (Fig. 6b; Supplementary Data 1). The range from the arabinosyl residue to the inositol was broader than that in Msm-Ara-LM with the most abundant products containing 15–24 mannosyl residues (thus with 14–23 residues between the arabinosyl and inositol units) (Supplementary Fig. 8b, d). Also, the initial mono-arabinosylated glycerolphosphoinositol mannans from *M. tuberculosis Cellulomonas*-digested LAM contained a broader and somewhat greater range of the number of mannosyl residues (Fig. 6b; Supplementary Fig. 8d).

**The presence of side chains containing multiple Man*p* units.** MS/MS analysis of the methylated *Cellulomonas*-treated *M. tuberculosis* LAM and Msm-Ara-LM molecules yielded unexpected ions corresponding to extensions of the side chains beyond a single mannosyl residue. In the case of Ara₁Man₂₄-Inos-P-Gro, the MS/MS spectrum of the triple- charged [M+ (NH₄)₃]³⁺ ion at *m/z* 1852.92, yielded a doubly charged (NH₄⁺)₂ Y ion at *m/z* 2661.30 (Fig. 7), which corresponds to Ara₁Man₂₃-Inos-P-Gro with a single OH group (see Figs. 3a and 8 for the formation of

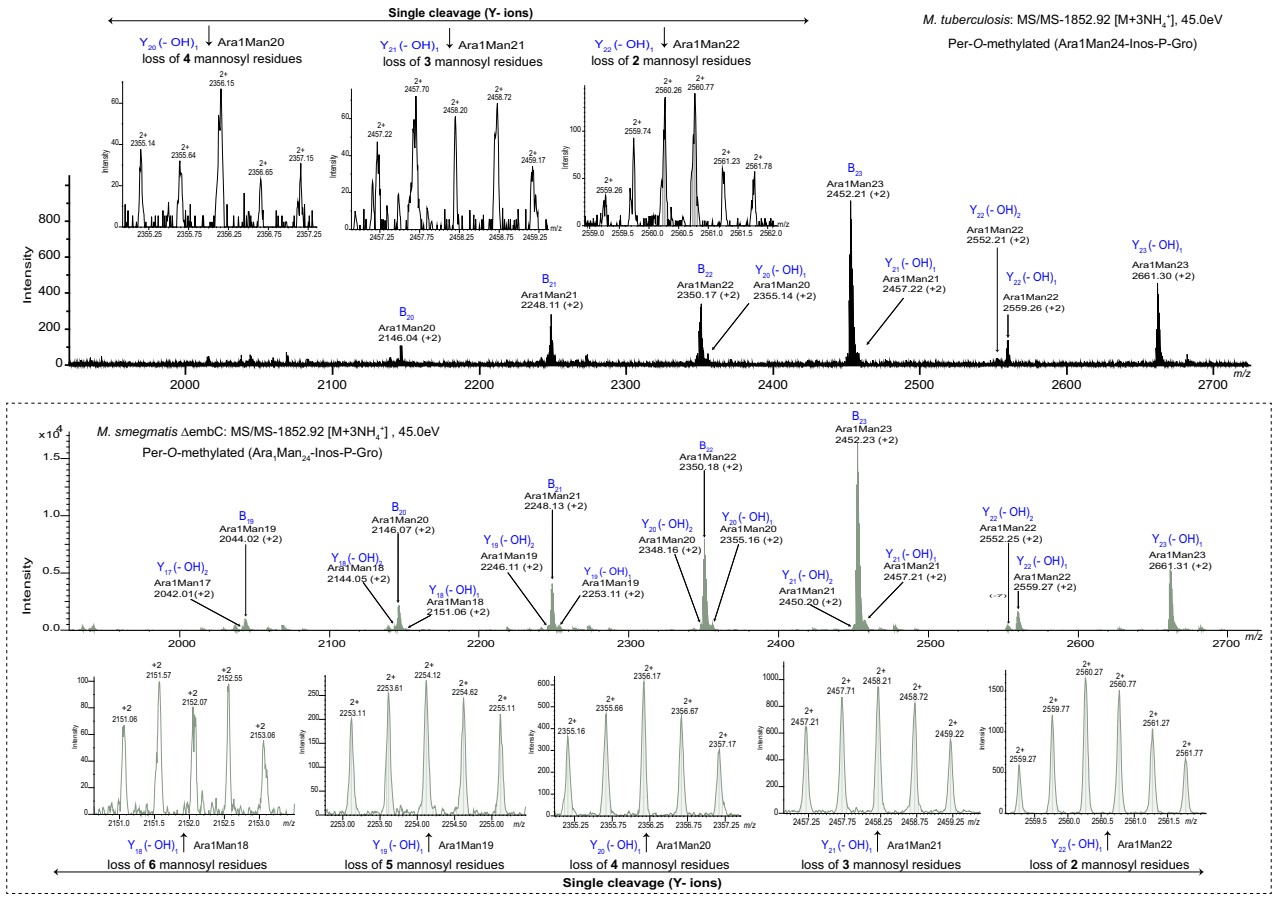

**Fig. 7 MS/MS fragmentation of per-*O*-methylated Ara₁Man₂₄-Inos-P-Gro in positive-ion mode.** A high-mass region of the MS/MS spectrum of the triple-charged [M + (NH₄)₃]³⁺ ion at *m/z* 1853 from **a** per-*O*-methylated Ara₁Man₂₄-Inos-P-Gro molecule produced by *Cellulomonas* endo-arabinanase treatment of *M. tuberculosis* LAM, and **b** per-*O*-methylated Ara₁Man₂₄-Inos-P-Gro molecule produced by *M. smegmatis ΔMSMEG_6387*. Although the B ions are not diagnostic, they are present in the region of the diagnostic Y ions that demonstrate the extended mannan backbone side chains and thus are labeled. The expansions in both **a**, **b** show the diagnostic Y ions in detail, which show the existence of secondary extended mannan side chains. In this figure, the *m/z* values of most intense peaks in any given isotope cluster are labeled, giving rise to a small discrepancy with the text where the *m/z* values of the ¹²C isotope peaks are given.

the Y ions). This ion is expected since it is formed from the loss of one of the unsubstituted terminal mannosyl units originally attached to O-2 of many of the 2,6-linked Man*p* units. It is to be expected that the loss of a second terminal mannosyl unit from a second 2,6-linked Man*p* residue (via a double cleavage) would also occur and yield a Y ion containing two OH groups at *m/z* 2552.21 due to the formation of two free hydroxyl groups, one from each mannosyl lost. As expected, this double cleavage does occur (Fig. 7). However, surprisingly, a clear and strong doubly charged (NH₄⁺)₂ Y ion with only one free OH, 7 *m/z* units higher (corresponding to 14 mass units) at *m/z* 2559.26 was also formed (Fig. 7) corresponding to the loss of two mannosyl residues by a single-cleavage event. For this to happen, the two Man*p* residues must be attached to each other. This same cleavage occurs down to the loss of six mannosyl residues by a single-cleavage event yielding Ara₁Man₁₈-Inos-P-Gro at *m/z* 2151.06 from *M. smegmatis* Ara₁Man₂₄-Inos-P-Gro (Fig. 7). However, the loss of seven mannosyl residues by a single cleavage occurs either very rarely or not since the ion at *m/z* 2049.00 is very weak or not present (Fig. 7). We then searched for B ions corresponding to the non-arabinosylated extended mannans and, as shown in Supplementary Fig. 9, ions resulting from unsubstituted oligomannans (i.e.,

mannans with no hydroxyl groups) with up to six mannosyl residues were found, although they were of low intensity.

The spectrum from the Ara₁Man₂₄-Inos-P-Gro from *M. tuberculosis* is similar to that from *M. smegmatis* Ara₁Man₂₄-Inos-P-Gro but weaker (Fig. 7). It yielded Y ions showing the loss of multiple Man*p* units down to the loss of four Man*p* units at *m/z* 2355.14.

Structures consistent with and illustrating these secondary backbone side chains are shown in Fig. 8. Similar Y-fragment ions were seen in the MS/MS spectrum of the triply charged (NH₄⁺)₃ ions for most other methylated Msm-Ara-LM species where MS/MS was performed including Ara₁-Man₂₁, ₂₃, ₂₅ Inos-P-Gro (data not shown). Since any given Ara₁Manₙ-Inos-P-Gro is a complex mixture of isomers, it cannot be concluded how frequently extended side chains occur, how many extended side chains are present, or if these extended side chains all have 6 mannosyl residues or whether some have a fewer number of mannosyl residues. What is clear is that the straightforward structure of the mannosyl backbone that always carries single α-Man residues is not always that case. Also, the extended branches provide a site for a second arabinosylation to the six-position of a mannosyl residue, raising the question of the structure of the infrequently

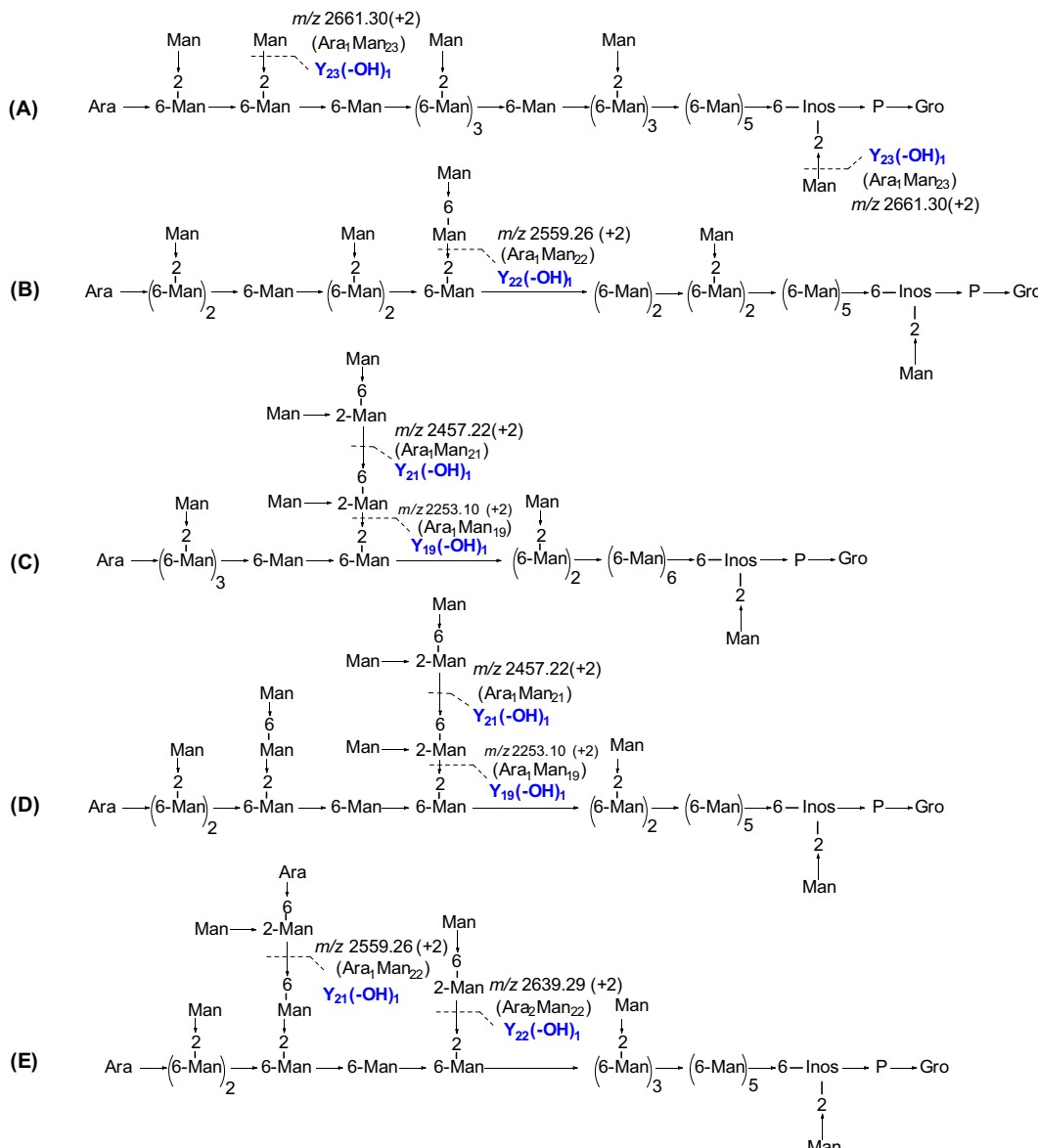

**Fig. 8 Five possible representative structures of the many structures consistent with the data herein of arabinosylated Man$_{24}$-Inos-P-Gro.** Structures **a**–**d** contain a single arabinosyl group; structure **e** contains two arabinosyl groups. Evidence for the nonbranched regions of 6-Man$p$ residues next to the inositol comes from earlier studies. Evidence for the preponderance of branched mannosyl residues next to the arabinosyl residue(s) comes from studies herein. Structure **a** containing no secondary branches. The arrangement of the branched and unbranched residues in the middle of the molecule is totally arbitrary. Structure **b** has a single extended secondary side chain of just one mannosyl residue. Its placement in the molecule is arbitrary. Structure **c** has a larger secondary side chain (5 mannosyl residues) and its placement is arbitrary; in this molecule, the Ara residue is only 13 mannosyl residues away from the inositol. Structure **d** has two extended secondary side chains of 2 and 3 mannosyl residues each, and again, as the extended secondary side chains increase either in size or number, the distance from the arabinosyl residue to inositol decreases. This leads us to believe that structures **a**, **b** may be more dominant. Finally, structure **e** has two arabinosyl residues with one residing on the main chain and one on an extended secondary side chain. The formation and $m/z$ values of some of the Y ions produced during MS/MS of the per-$O$-methylated molecules (Figs. 7, 9) are illustrated. Other Y ions present in Figs. 7, 9 come from structures not illustrated in Fig. 8.

occurring Ara$_2$LM molecules produced by *M. smegmatis* $\Delta$MSMEG_6387.

**Evidence for two arabinosyl residues attached to the LM backbone at two different sites.** Two explanations for the quantitatively minor Ara$_2$LM species (see Fig. 2c) are possible. In one possibility, two arabinosyl units are attached to each other due to some residual α-1,5 arabinosyltransferase activity active on Msm-Ara$_1$-LM in the *embC* knockout. In the second possibility, the two arabinosyl residues are attached to the LM unit in two

different places as shown in Fig. 8e. The first possibility is very likely for the Ara$_2$-LM species from *M. tuberculosis* LAM (where EmbC is active) that was digested with *Cellulomonas* endo-arabinanase, as the endo-arabinanase is likely to leave a "stub" of α-Ara$f$-(1→5)-α-Ara$f$. Indeed this "stub" was shown to be present in the isolated tetrasaccharide Ara$f$-(1→5)-Ara$f$-(1→6)-Man$p$-(1→6)-Man (Fig. 3c).

For further structural analysis of two arabinosyl residues on the mannan backbone, the per-$O$-methylated Ara$_2$Man$_{24}$-Inos-P-Gro was identified by LC–MS as a triply charged $[M + (NH_4)_3]^{3+}$ ion at $m/z$ 1906.27 (Supplementary Fig. 10). MS/MS analysis of the

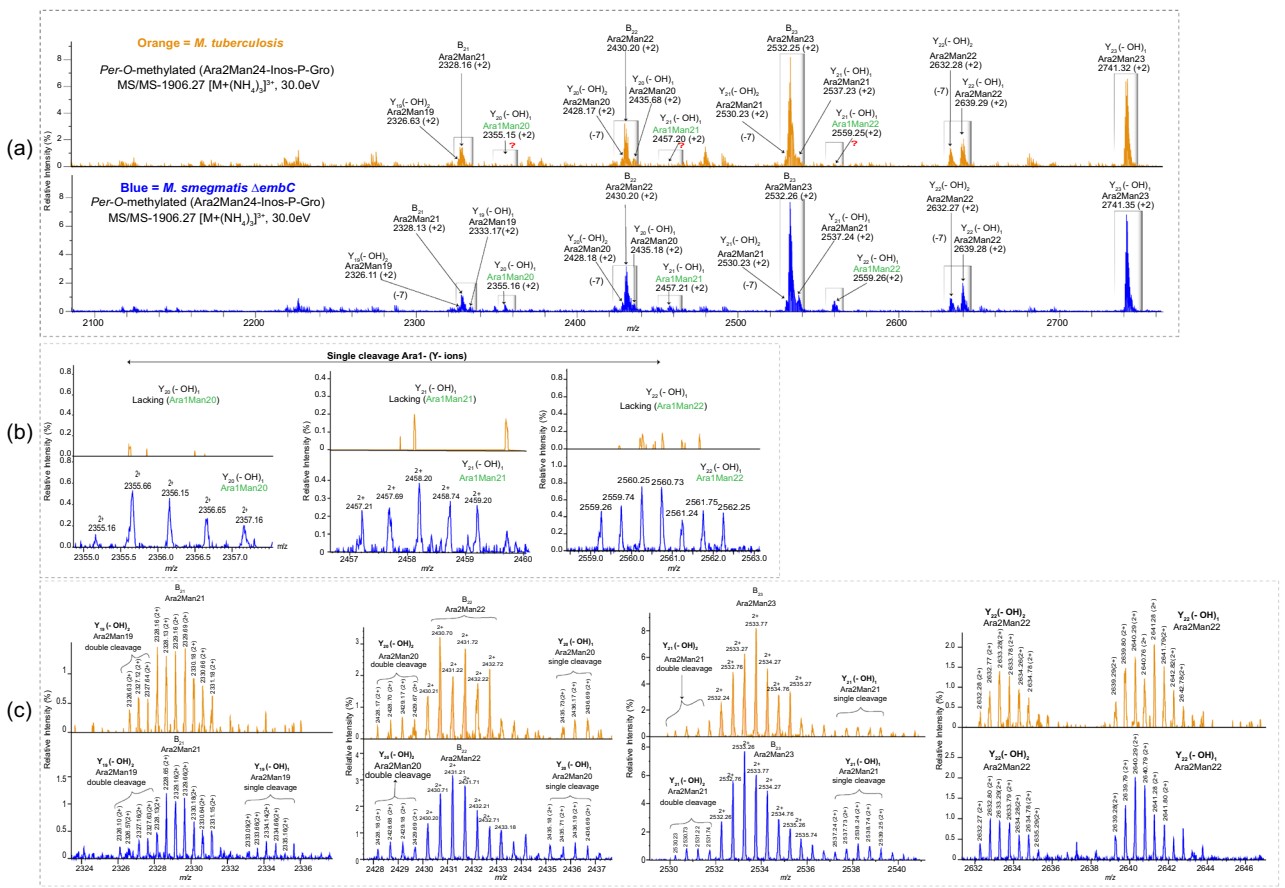

**Fig. 9 MS/MS fragmentation of per-O-methylated Ara$_2$Man$_{24}$-Inos-P-Gro in positive-ion mode. a** High-mass region of the MS/MS spectra of the triple-charged $[M + (NH_4)_3]^{3+}$ ions at $m/z$ 1906.27 from the per-O-methylated Ara$_2$Man$_{24}$-Inos-P-Gro molecule produced by *Cellulomonas*-digested *M. tuberculosis* LAM (orange) and *M. smegmatis* Δ*MSMEG_6387* (blue). **b** Y ions with one arabinose, which are present in the *M. smegmatis* Δ*MSMEG_6387* (blue) and absent in the *Cellulomonas*-digested *M. tuberculosis* LAM (orange) showing that the two arabinosyl residues are in separate regions on the *M. smegmatis* Δ*MSMEG_6387* (blue). **c** Y ions with two arabinosyl residues present in both molecules, which show that the Ara$_2$Man$_{24}$-lnos-P-Gro molecules also contain extended secondary mannan side chains. The base peak chromatograms and MS[1] spectra are shown in Supplementary Fig. 10.

per-O-methylated *M. smegmatis* Ara$_2$Man$_{24}$-Inos-P-Gro was suggestive that at least some of the isomers consisted of arabinosyl substitution at two different places in the molecule, as revealed by an analysis of the doubly charged $(NH_4^+)_2$ Y ions (Fig. 9a, b). Thus, as expected, a strong Y ion formed by the loss of a single mannosyl residue with a composition of Ara$_2$Man$_{23}$-Inos-P-Gro at $m/z$ 2741.35 was present. Also, as in the Ara$_1$-Man$_{24}$-Inos-P-Gro species, loss of multiple mannosyl residues by a single cleavage was seen at $m/z$'s 2639.28, 2537.24, 2435.18, and 2333.17 corresponding to Ara$_2$Man$_{22–19}$-Inos-P-Gro being formed by a single cleavage for losses of up to 5 mannosyl residues (Fig. 9c). However, ions were also seen at $m/z$'s 2559.26, 2457.21, and 2355.16 (Fig. 9b), and extremely weak ions at $m/z$'s 2253.10 and 2151.06, which correspond to the doubly charged Y ions for Ara$_1$Man$_{22–18}$-Inos-P-Gro resulting from losses of Ara$_1$Man$_{2–6}$. The mass accuracy values for these corresponding fragment ions are given in Supplementary Table 4. We thus conclude that at least some of the Ara$_2$Man$_{24}$-Inos-P-Gro molecules have arabinosyl residues at two different positions although it remains possible that some of the molecules also have a disaccharide of arabinosyl residues as this analysis would not necessarily have detected them if they were weak. We must now modify that claim in our earlier paper[24] that only a single arabinan chain is present in LAM with the conclusion that mostly a single arabinan chain is present, but di- and possibly tri-arabinosylated mannans exist. As shown in Fig. 9b, in the *M.*

*tuberculosis* Ara$_2$Man$_{24}$-Inos-P-Gro MS/MS spectrum, ions at $m/z$'s 2559.26, 2457.21, and 2355.16 are not present. This is as expected because the Ara$_2$Man$_{24}$-Inos-P-Gro is a product of incomplete digestion of the *Cellulomonas* endo-arabinanase, and mostly the Ara$f$ residues are attached to each other (we cannot rule out small amounts of arabinosyl residues at two different positions in *M. tuberculosis* LAM that did not show up in the mass spectrum). The lack of these Y ions also supports the contention that the Y ions used to show the existence of extended mannan side chains are correctly interpreted.

## Discussion

To isolate an oligosaccharide containing Ara$f$ and Man$p$, we began with Msm-Ara-LM from *M. smegmatis* Δ*MSMEG_6387* that we have previously shown predominately produces LM substituted with a single Ara$f$ residue[24] due to the lack of an α-1,5 arabinofuranosyl transferase. With our techniques worked out to isolate and characterize the oligosaccharide(s) containing Ara$f$ and Man$p$, we then turned to *M. tuberculosis* LAM that had been treated with *Cellulomonas* endo-arabinanase[30] that was more heterogeneous with respect to the number of arabinosyl residues attached to the LM core due to the action of the endo-arabinanase. Since Ara$f$ residues are more readily hydrolyzed than Man$p$ residues, it required an enzymatic procedure to isolate oligosaccharides containing both arabinofuranosyl and manno-pyranosyl residues from these sources. Thus, critical to this

project was obtaining the 1,6-α-endo-mannanase first character-ized by Clinton Ballou over 40 years ago but no longer readily available. We chose to clone and express the gene from *Bacillus circulans* encoding this enzyme, and a detailed report of this work has recently been published[28]. In order for the enzyme to yield small oligosaccharides, we incubated the arabinosylated LM with α-mannosidase, which is available commercially thus removing all mannosyl branches and simplifying the endo-mannanase substrate. A single trisaccharide, Ara-Man-Man, was obtained from *M. smegmatis*. Care was taken to positively identify the linkages present and, thus, although the MS/MS spectra (Fig. 3) showed that both the interior and reducing end residues were six-linked, we went on to purify the reduced and methylated oligo-saccharide from *M. smegmatis*, and convert it into alditol acetates to show unequivocally that both the interior hexosyl residue and the reducing end hexitol residue were 6-linked, and the arabinosyl residue was furanosyl (Supplementary Fig. 4) and thus yielding the structure Ara$f$-(1→6)-Hex$p$-(1→6)-Hex. Since small amounts of glucose are always present in Msm-Ara-LM preparation, we also took care to show by retention times that these six carbon residues were both in the *manno* and not *gluco* configuration (Supplementary Fig. 4). This result conflicted with our earlier report[17]; we now recognize that at that time, we neglected to take into account that although Ara$f$ residues are more readily hydrolyzed than Man$p$ residues, the fact that there are over tenfold more Man$p$ than Ara$f$ residues greatly diminishes the specificity of the partial acid hydrolysis experiment. We went on to show by MS/MS of the methylated Msm-Ara-LM that both of the mannosyl residues of the trisaccharide were substituted with single Man$p$ residues at O-2 before α-mannosidase treatment, a result consistent with the data reported in our earlier study[17], which, upon reinterpretation, showed that the Man residues to which Ara was attached were 2,6-linked.

We showed that the exact same trisaccharide Ara$f$-(1→6)-Man$p$-(1→6)-Man was present in *M. tuberculosis* LAM as well as the tetrasaccharide Ara$f$-(1→5Ara$f$-(1→6)-Man$p$-(1→6)-Man. The isolation and characterization of the tetrasaccharide showed that the penultimate reducing end arabinosyl residue is attached to position 5 of the reducing end Ara$f$, which is not surprising, and probably is attached by the α-1,5 arabinosyltransferase EmbC in the wild-type *M. tuberculosis* LAM, and is missing in *M. smegmatis ΔMSMEG_6387* in which EmbC is not functional.

We then asked the question of how many mannosyl residues (whether or not they were substituted at O-2) are between the T-Ara$f$ and the inositol unit by analyzing Msm-Ara-LM that had been treated with α-mannosidase. Analysis of the pseudomole-cular ions revealed that the dominant number of mannosyl residues between the T-Ara$f$ residue and inositol was 13–18 for Msm-Ara-LM and 14–23 for *M. tuberculosis* LAM (Fig. 6; Supplementary Fig. 8).

This result was consistent with the MALDI–TOF analysis of *M. smegmatis ΔMSMEG_4247* that lacks the branching mannosyl enzyme and produces "LAM" with a linear α-1,6 mannan back-bone substituted with a "normal" arabinan[31]. Removal of the majority of the arabinosyl residues of this product with *Cellulomonas* endo-arabinanase yielded molecules with a single Ara residue with 10–18 mannosyl residues, among these 15–17 mannosyl residues being dominant.

When analyzing the MS/MS spectrum of methylated Msm-Ara-LM, we were surprised to find Ara-containing Y ions pro-duced by the loss of multiple mannosyl ions in a single-cleavage event (Fig. 7). We confirmed these secondary side chains by observing the presence of the corresponding B ions (Supple-mentary Fig. 10). Similar ions were found in the MS/MS spectrum of methylated *M. tuberculosis* Cellulomonas endo-arabinanase-treated LAM (Fig. 7), and indeed in the MS/MS spectrum of

methylated Ara$_2$Man$_{24}$-Inos-P-Gro from both species (Fig. 9). Although this mass spectral evidence from all these angles is rather convincing as to the existence of the extended side chains, it is an unexpected result, and it remains important to show the presence of these secondary side chains by other methods as their presence profoundly changes our understanding of a simple mannan backbone structure. It is not surprising that the NMR, acetolysis, and methylation analyses used previously to determine the structure of the mannan backbone did not detect the sec-ondary side chains since little or no changes in the results of these analyses would be expected whether extended side chains are present or not. In the MS methods used here, the arabinosyl residue was a critical marker of the main chain; similar MS analysis of non-arabinosylated LM would not have allowed Y ions produced from the main chain to be distinguished from Y ions produced from the secondary extended side chains as both would have the same composition and thus give the same *m/z* values.

An attempt to illustrate a few of the possible isomers present is shown for arabinosylated Man$_{24}$-Inos-P-Gro in Fig. 8. However, it must be emphasized that much remains to be deciphered about the structure of the secondary side chains. It is not known if they come in various sizes from 2 to 6 mannosyl residues. It is also not yet known how commonly these side chains occur. The distance between the arabinosyl residue and the inositol as well as the glycosyl linkage composition all suggest fewer and shorter sec-ondary extended side chains and the presence of molecules with no secondary extended side chains at all (Fig. 8a). Finally, the positions of the secondary extended side chains along the primary mannan backbone remain to be elucidated.

The extended mannosyl side chains give rise to additional non-reducing ends to which an arabinosyl residue can be attached and, indeed, we showed that in Msm-Ara-LM, a small percentage of the molecules is arabinosylated in two different places on the mannan (Figs. 9 and 8e). In the case of the *M. tuberculosis* LAM, the presence of large amounts of Ara$f$-(1→5Ara$f$) "stubs" attached to the mannan were produced by the *Cellulomonas* digestion precluded obtaining mass spectral evidence for arabi-nosylation occurring in two different places on the mannan backbone. However, the presence of extended mannosyl side chains on the *M. tuberculosis* mannans (Fig. 7) is suggestive that arabinosylation in two different places on the mannan may likely occur in *M. tuberculosis* LAM as well as in *M. smegmatis* LAM.

Collectively, these findings call for a significant revision of the structure of LAM, and also raise a number of biosynthetic questions. Most straightforwardly, the order of addition of the Ara$f$ and the Man$p$ residues around the Ara$f$ residue is not clear. The fact that the disaccharide α-Man$p$-(1→6)-α-Man-octyl is arabinosylated by *M. smegmatis* enzymes[25] suggests that arabi-nosylation can occur before mannosylation at O-2, but does not rule out the possibility that the arabinosyltransferase is indifferent to mannosyl residues at O-2 of the mannosyl receptor. Nor is it clear whether mannosylation at O-2 of the arabinosylated man-nosyl residue can occur by the branching enzyme. Thus, in vivo, there might be a strict order addition of glycosyl residues to form the dominant structure around the arabinosyl residue illustrated in all structures in Fig. 8. The data by the Morita group[32] that overexpression of the branching enzyme reduces the length of the mannan in both LM and LAM might suggest that arabinosylation of a terminal two-linked mannosyl residue at the six-position can occur, but mannosylation of such a residue to further elongate the mannan does not occur. In vitro, arabinosyltransferase assays with various mannosyl acceptors should shed some light on the biosynthetic pathway.

A more fundamental biosynthetic question concerns the enzymes involved in forming the extended side chains. Can the extended side chain be synthesized by the same enzymes that

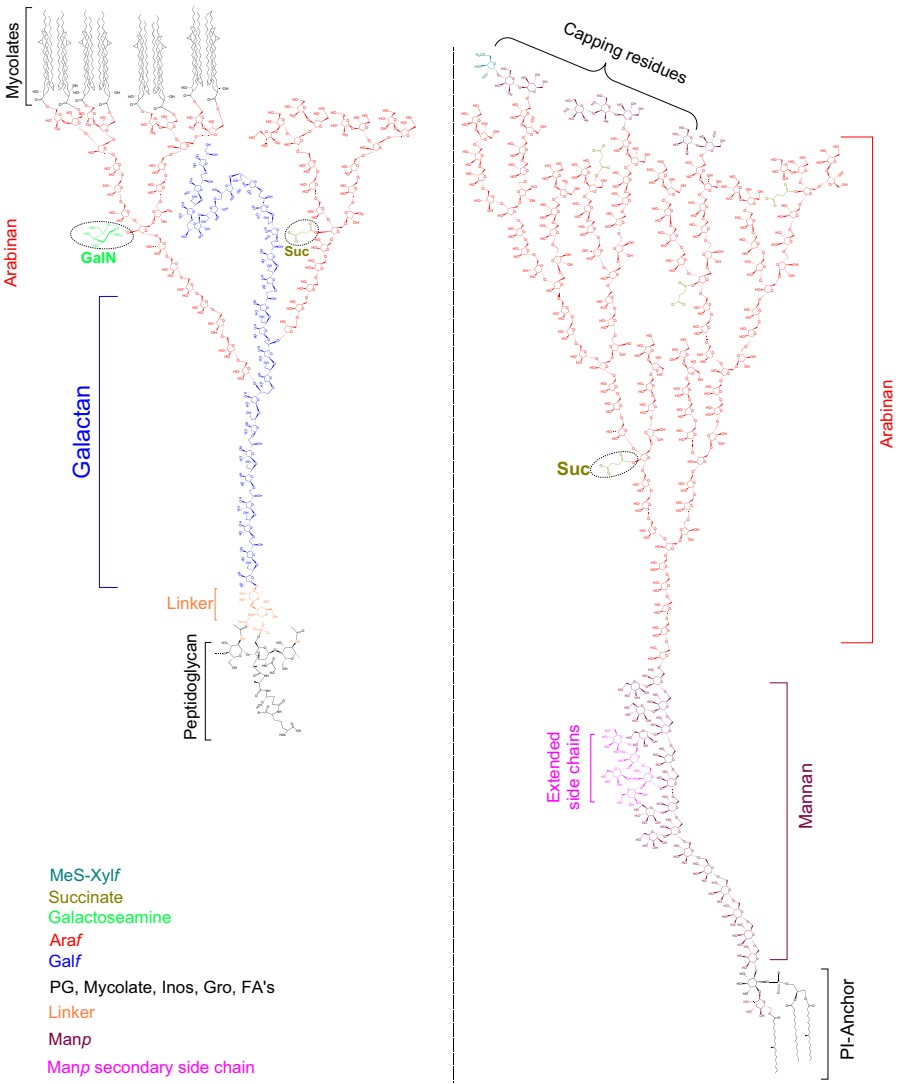

MeS-Xyl*f*
Succinate
Galactoseamine
Ara*f*
Gal*f*
PG, Mycolate, Inos, Gro, FA's
Linker
Man*p*
Man*p* secondary side chain

**Fig. 10 Current structural models that contrast arabinogalactan and mannose-capped lipoarabinomannan from *M. tuberculosis* H37Rv. a** Arabinogalactan (AG). **b** Mannose‑capped lipoarabinomannan (ManLAM). The mannan in ManLAM is that of Fig. 8c. The primary structures for the arabinan models for AG[39] and LAM[20] are consistent with our enzymatic degradation data. For LAM, only a single arabinan chain is attached to the mannan as shown to be most dominant herein and previously[24]. The two-dimensional arrangements of the glycosyl residues shown are reasonable but not based on any data, and some liberties with bond lengths and angles have been taken for purposes of legibility.

biosynthesize the main backbone chain, or are some of the other glycosyltransferases present in the genome but with no known function involved?

From a host–pathogen interaction standpoint, it is well-established that subtle variations in the structures of LM and LAM, including the length and branching of their mannan backbone, dramatically impact their biological activities[33–35]. The revised structure of the mannan backbone of LAM presented herein provides opportunities to revisit the specific impact of this region of the lipoglycan on immunomodulation (e.g., using more physiologically relevant mannan backbone mimetics), and to explore the impact of extended mannan side chains on these activities.

Finally, it is worth contrasting the structures and functions of LAM and AG as they are currently understood (Fig. 10). The function of AG is primarily structural in tethering the pepti-doglycan to the outer "mycolyl" membrane. Hence, its length and glycosyl arrangement must be consistent with that end. The galactan and linker region (except for the attachment of the arabinans) are linear and unbranched. The arabinan contains an unbranched region of α-5-linked Ara*f* residues containing perhaps seven residues (this may vary within limits), which are then attached to a defined branched structure of 17 arabinosyl units. Thus, although there is room for some variation, the primary structure of AG is constrained to fit its function. The function of LAM is less straightforward. Clearly, it is active immunologically, and its various biological activities are dependent on the fine details of its structure[3–5]. Its physiological function in the mycobacterial cell membranes, however, is less clear. The structural studies reveal a structure that is more highly branched and more heterogeneous than AG. This is especially true of the arabinan[20]. The mannan is only non-branched near the inositol end and otherwise branched with both single α-Man*p* residues and, as shown herein, with extended Man*p* oligosaccharides. Further, as revealed by the α-mannosidase products, the attachment of the arabinan occurs on mannans of different lengths. It is likely that all this het-erogeneity and complexity of branching of both the mannan and arabinan has physiological and/or immunomodulatory purposes, whether in the periplasmic space (for LAM anchored

in the plasma membrane) or at the cell surface of the bacteria (for LAM anchored in the outer membrane).

## Methods

**Materials**. *M. smegmatis* Ara-LM was produced and purified as previously described[19]. Purified *M. tuberculosis* LAM was obtained from BEI resources. Endo α-1,6-mannanase was prepared as described[28]. Alpha-mannosidase (Jack Bean) was obtained from Sigma/Aldrich. *Cellulomonas* endo-arabinanase[30] was kindly provided by Dr. Delphi Chatterjee and Anita Amin (Colorado State University).

**Liquid chromatography–mass spectrometry**. Methylated and acetylated trisaccharides from *M. smegmatis* ΔMSMEG_6387 Ara-LM were analyzed using an Agilent 1200 series high-performance liquid chromatography (HPLC) binary pump coupled with an Agilent 6220A-TOF mass spectrometer. HPLC chromatographic separation was carried out using Waters reverse-phase X-bridge C18 column (150 mm × 2.1 mm; 1.7 μM) with a flow rate of 0.32 ml/min using solvent A (0.1% formic acid in water) and solvent B (0.1% formic acid in methanol). Typically, analytes 100 ng/ul (assuming all the derivatized material recovered) were injected in 25% solvent B (75% Solvent A) followed by a 30-min gradient to 100% solvent B, which was further held for an additional 10 min. For isolation of pre-reduced, per-O-methylated Araf-(1→6)-Manp-(1→6)-mannitol from the enzymatic digestion mixture, a modified HPLC chromatograpic separation method was adopted by maintaining an isocratic condition using 60% solvent B (0.1% formic acid in methanol) for 5 min. Further, a similar isocratic gradient consisting of 98% solvent A with 10 mM ammonium acetate for 5 min was used to isolate *Cellulomonas* endo-arabinanase-digested *M. tuberculosis* LAM from other digested arabinan fragments. Isocratic separations were performed with the same column noted above.

The mass spectrometer (Agilent 6220A-TOF) was equipped with a combination electrospray ionization/atmospheric pressure chemical ionization (ESI/APCI) multimode source operated in positive-ion mode. The mass spectra were recorded at a rate of 1.02 spectra/s with a data acquisition range of m/z 250–3200 Da. Data processing was carried out using Mass Hunter Workstation Software Qualitative Analysis (version B.07.00).

**LC-MS/MS of trisaccharide from *M. smegmatis* ΔMSMEG_6387**. The above LC-purified, prereduced, per-O-methylated Araf-(1→6)-Manp-(1→6)-mannitol sample was separated using a Waters UPLC system with a Waters T3 C18 UPLC column (1 × 100 mm, 1.8 μM). Mobile phase A consisted of water + 0.1% formic acid, and mobile phase B consisted of acetonitrile + 0.1% formic acid. Flow rate was held constant at 0.2 mL/min, and the column temperature was set to 50 °C. The gradient consisted of 99.9% A for 1 min, a linear ramp to 99.9% B over 12 min, a 3-min hold at 99.9% B, and a return to starting conditions over 0.05 min with 3.95 min of equilibration. For MS/MS analysis, the UPLC eluent was directed to a Waters Xevo G2 Q-TOF running in positive-ionization mode. Mass was calibrated using sodium formate with less than 1-ppm mass error. LockMass reference of leucine enkaphalin was used to ensure mass accuracy through the run. The electrospray source conditions included capillary voltage of 2.2 kV, cone voltage = 30 V, extraction cone = 4 V, temperature = 150 °C, desolvation temperature of 350 °C, and desolvation gas flow of 800 L/h. Data were acquired in centroid mode from m/z 50–1200 using 0.2-s scan times. Collision energy 45 V was identified optimal to induce MS/MS fragmentation. Data analysis were performed using MassLynx V4. 1 SCN803 (Waters) software.

**Gas chromatography–mass spectrometry**. To further resolve the linkage analysis of the LC-MS-purified Araf-(1→6)-Manp-(1→6)-mannitol, the dried trisaccharide (10 μg) was hydrolyzed in 2 M TFA, 120 °C for 2 h. After drying under N2, it was reduced for 2 h with NaB[²H]₄, acetylated, and analyzed by gas chromatography–mass spectrometry (GC–MS). GC–MS analysis of permethylated alditol acetates was carried out using TRACE 1310 Gas Chromatograph (Thermo Scientific) equipped with a TSQ 8000 Evo triple-quad GC–MS/MS (Thermo Scientific). GC was connected to ZB-5HT Inferno capillary column (30 m × 0.25 mm i.d × 0.25-μm thickness, Phenomenex, CA, USA). The samples were injected (50 ng/μl, assuming all the derivatized material recovered) in the split-less mode at an initial temperature of 100 °C, under the constant flow of Helium carrier gas at a rate of 5 ml/min. Oven temperature gradient profile: begins at 100 °C, +20 °C/min to 150 °C, +5 °C/min to 240 °C (hold 3 min), and +30 °C/min to 300 °C (hold 5 min), with total acquisition time of 30 min. The mass spectrum was scanned from m/z 50–500; data analysis was performed using Chromeleon Chromatography data system software (Thermo Scientific).

LC separation and mass spectrometry of intact native *M. smegmatis* ΔMSMEG_6387 deacylated Ara-LM, *Cellulomonas* endo-arabinanase-digested *M. tuberculosis* LAM, and α-mannosidase-treated materials were performed on a Waters ACQUITY ultra-performance liquid chromatography (UPLC) instrument. UPLC separations were performed using reverse-phase X-bridge C18 column, 50 mm × 2.1 mm, 1.7 μM, using a linear gradient of 10–70% acetonitrile at 0.4 mL/min in 6.8 min. The injection volume was set to 200 ng/μl. The mobile phase was composed of solvent A (water), solvent B (acetonitrile), and solvent C (500 mM ammonium acetate); 2% (v/v) was maintained throughout the chromatographic

separation for a final concentration of 10 mM ammonium acetate in the gradient conditions. A modified gradient of 30–100% acetonitrile was used to identify per-O-methylated polysaccharides.

High-resolution mass spectrometry was performed on a Bruker maxis plus II quadrupole TOF (Q-TOF) instrument. Native samples were dissolved in 10 mM ammonium acetate, whereas derivatized (per-OO-methylated) samples were dissolved in methanol. The ESI source parameters (for positive and negative modes) were as follows: end plate offset voltage 500 V, capillary voltage 3500 V, nebulizer gas pressure 3.0 bar, and drying gas flow rate 10 L/min. Two different tune parameters were used for analyzing native and derivatized products. Given below are the tune parameters used to detect low molecular compounds (≤m/z 1200 Da): funnel RF 300 Vpp, multipole RF 300 Vpp, ion energy 3.0 eV, low-mass range for ion transmission was set to m/z 100, collision energy 8.0 eV, collision RF 450 Vpp, prepulse storage 5.0 μs, ion cooler RF 800 Vpp, and transfer time 80.0 μs. Modified tune parameters were employed to detect high-molecular-weight compounds (≤m/z 10,000 Da) as follows: funnel RF 400 Vpp, multipole RF 400 Vpp, ion energy 3.0 eV, low-mass range for ion transmission m/z 600, collision energy 8.0 eV, collision RF 2500 Vpp, prepulse storage 5.0 μs, ion cooler RF 800 Vpp, and transfer time 140 μs.

The MS and MS/MS spectra for native material were obtained in negative mode, whereas, the per-O-methylated samples were collected in positive mode. High-energy (70, 80, and 90 eV) collision-induced dissociation (CID) was used to induce MS/MS fragmentation of underivatized polysaccharides. Derivatized (per-O-methylated) polysaccharides required low CID energies. The precursor ion isolation width for MS/MS experiments was set to ±4 m/z. Mass spectrum data analysis was performed using Bruker compass data analysis 4.4 SR1.

**Preparation of *M. smegmatis* ΔMSMEG_6387 Ara-LM and digestion of Msm-Ara-LM with degradative enzymes and purification of Araf-(1→6)-Manp-(1→6)-Man**. Sephacryl S-200/S-100 column (Amersham Biosciences) purified Ara-LM from *M. smegmatis* ΔMSMEG_6387 as previously described[24] was further subjected to additional hydrophobic interaction column chromatography using octyl-Sepharose (Pharmacia Biotech Inc.) to remove free neutral glycans[36]. Purified Ara-LM obtained (5 mg) was deacylated with 0.2 N NaOH at 37 °C for 2 h. The mixture was neutralized with (10% aqueous acetic acid) and applied to a column (1 × 50 cm) of Bio-Gel P4 resin and eluted with water. The entire deacylated Ara-LM obtained was then digested with 100 μL of α-mannosidase (15 units/mg protein, Sigma) in a total volume of 1 ml of sodium acetate buffer (0.05 M, pH 4.5) containing 0.02% NaN₃ and incubated at 37 °C for 16 h. After overnight digestion, another 100 μL of α-mannosidase was added and continued for further 8 h. Enzymatic activity was heat-inactivated, and the digested sample was applied to P4 resin and eluted with water. The water-eluted fractions containing carbohydrate fractions were monitored by thin-layer chromatography. Pooled α-mannosidase-digested Msm-Ara-LM was further digested with endo-1,6-mannanase (GH-emn) as previously described[17] and desalted on a P4 column. The water-eluted fractions were spotted on TLC to test for the presence of carbohydrates. An aliquot of TLC-positive fractions (5% of each fraction volume) were further acetylated to identify the Araf-(1→6)-Manp-(1→6)-Man (trisaccharide) by LC–MS analysis as described above for the Agilent instrument. The LC–MS-identified Araf-(1→6)-Manp-(1→6)-Man-containing fractions were pooled, dried, and reduced overnight with NaB[²H]₄, and the resulting oligoglycosylalditols were per-O-methylated as described[37]. Prereduced, methylated Araf-(1→6)-Manp-(1→6)-mannitol was further isolated from other digested products by liquid chromatographic isocratic gradient method as described above for the Agilent instrument.

*M. tuberculosis* LAM obtained from BEI resources (2 mg) was deacylated with 0.2 N NaOH at 37 °C for 2 h and then neutralized with 10% aqueous glacial acetic acid. Deacylated *M. tuberculosis* LAM (d-LAM) was further separated from sodium salts using Amicon ultra-0.5 mL centrifugal filter (3-kDa MWCO). The d-LAM was then digested with *Cellulomonas* endo-arabinanase as previously described[38] and purified from endo-arabinanase-released arabinan fragments by the LC–MS method described above for the Agilent instrument. This material was further analyzed on Bruker maxi plus II Q-TOF instrument to understand the heterogeneous population of LM backbone with varying number of arabinosyl residues (0–3).

The *Cellulomonas*-digested *M. tuberculosis* d-LAM (50 μg) was digested directly in LC–MS sample vials with 50 μL of α-mannosidase (15 units/mg protein, Sigma) in a total volume of 200 μl in 10 mM ammonium acetate buffer (pH 6). The digestion mixture was incubated at 37 °C for 24 h and analyzed on the Bruker maxi plus II Q-TOF instrument without any further purification. After 24 h of α-mannosidase digestion, samples were heat-inactivated and further digested in the same buffer with endo-1,6-mannanase (GH-emn) as previously described[28]. The native underivatized enzymatic digested products containing Araf-(1→6)-Manp-(1→6)-Manp (trisaccharide) and Araf-(1→5)-Araf-(1→6)-Manp-(1→6)-Manp (tetrasaccharide) were identified by LC–MS analysis using Bruker maxi plus II Q-TOF instrument, and MS/MS fragmentation was obtained at collision energy of 25.0 eV in negative mode using low-mass tune parameters.

**Methylation of poly- and oligosaccharides**. For the purpose of methylation only, LAM (100 μg) from both sources *M. tuberculosis* and *M. smegmatis* ΔMSMEG_6387 was per-O-methylated using the modified procedure[37].

Methylated samples were analyzed on Bruker maxi plus II Q-TOF instrument in positive mode. Collision energy for MS/MS fragmentation was optimized between 20 and 45 eV using high and low-mass tune parameters.

## Data availability

The data that support the findings of this study are available from the corresponding author upon reasonable request.

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

## Acknowledgements

The authors gratefully acknowledge the preparation of the *Cellulomonas* endo-arabinanase enzyme by Anita Amin and discussions with Drs. Delphi Chatterjee and De Prithwiraj. We thank Michael Scherman for helping with the drawings in Fig. 10. We thank Dr. Michael Lyons for writing an algorithm to convert the intensity of the $^{12}C$-only isotope component into the intensity of the $^{12}C$-only isotope component plus all the $^{13}C$-containing components as reported in Supplementary Data 1 and Fig. 6. We also thank Megan Lucas for assisting in the LC purifications, Zachary Moen for running LC–MS samples, and Corey Broeckling for running LC–MS/MS on the per-*O*-methylated Ara*f*-(1→6)-Man*p*-(1→6)-mannitol sample. The following reagent was obtained through BEI Resources, NIAID, NIH: *M. tuberculosis* strain H37Rv, Lipoarabinomannan (LAM) NR-14848 (lot 62414040). This work was supported by the National Institutes of Health/

National Institute of Allergy and Infectious Diseases grant AI064798. The content is solely the responsibility of the authors and does not necessarily represent the official views of the NIH.

## Author contributions

M.J., M.R.M., and S.A. designed the research; S.A. performed the experiments; W.L. overproduced and purified endo-mannanase; M.R.M. and S.A. analyzed the data; C.M.B. designed and oversaw the LC–MS and LC–MS/MS experiments on the Maxis plus II (Bruker) instrument; M.J., M.R.M., and S.A. wrote the paper.

## Competing interests

The authors declare no competing interests.
