## [Peer Review File · Communications Chemistry]

Reviewers' comments:

Reviewer #1 (Remarks to the Author):

In this work, Angala and co-workers, determine that the arabinan component of the mycobacterial lipoarabinomannan (LAM) is attached at the nonreducing end of the mannan rather than to internal regions. In addition, the authors identified the presence of secondary extended mannan side chains attached to the internal mannan region, leading not only to a significant revision of the chemical structure of LAM, but also to the structural and functional role of the lypoglycan both in the periplasm and outside the cell envelope. The manuscript is clear and well written. Because of its novelty and the quality of the work, I support the publication of this manuscript in Communications Chemistry.

Reviewer #2 (Remarks to the Author):

The manuscript entitled "The Presence of Secondary Extended Mannan Side Chains and the Attachment of the Arabinan in Mycobacterial Lipoarabinomannan" by Angala et al. it is an expansion of a previous report describing how the arabinan domain of the lipoarabinomannan molecule attaches to the mannan domain. In addition, it also describes the novel finding of the existence of a secondary extended branched pentasaccharide mannose branch from the mannan-core. This an interesting report revisiting the structure of LAM and LM from a very well established group in the biosynthesis of the mycobacterial cell wall. There are some suggestions that will excel the quality of this reported study:

It is not clear how many times these experiments were performed and how data are being reported (means?). And if it is a mean value, where are the SD or SEM? This is obvious in Fig 5, where it is not clear how many times this has been done and if the digestion was exhaustive or not.

Some of the m/z values mentioned in the text do not correspond to the exact values in the figure, please revise (page 8 reports m/z 2661.32 but in figure is m/z 2262.32), the gain/loss of a proton needs to be accounted. If this is not the case, clarify why.

Revise Fig 7c, if I am not mistaken, the m/z value depicted for Y21 (-OH)₁ should be m/z 2458.21 in place of m/z 2559.

It will be interesting to determine or comment if specific acylations (e.g. succinates) indicate stooping points or branching for mannosyl- and arabinosyl-transferases.

In the discussion, the authors need to expand about the biological meaning of their results, what these may mean knowing how *M. tuberculosis* uses this molecule as a virulent factor the gain entry and survive in host cells.

Does the LAM heterogeneity only has to do with different stages of LAM production at the time that LAM is being purified or is the LAM structure dynamic?

The last sentence in the discussion: To which purpose?

The blue labeling in Fig 1 as depicted may be confusing to the general reader, as increments do not seem to correspond to a hexose. Clarify in the figure legend.

Fig 2c needs to be referenced in the text.

Fig 9 specify from which mycobacterial species this LAM structure is from, not PI-caps or Man-caps are not depicted.

Consider to change: This data suggests... to These data suggest...

Reviewer #3 (Remarks to the Author):

This paper uses high resolution mass spectrometry to characterize lipoarabinomannan, an arabinosylated lipopolysaccharide of mycobacteria. The authors identify previously undiscovered mannan side chains, and utilize various enzymatic procedures to help map the composition of the lipoarabinomannan. The results have the potential to be exciting to a certain part of the bio community; however, the current state of the manuscript falls short in the quality of the figures, the lack of rigor in the supporting information, and in some cases the description of the data and interpretation. Moreover, although the depth of analysis is impressive, the impact of the paper is overshadowed by the focus on extreme data analysis coupled with lack of appropriate supporting data.

On page 2, the authors indicate "LAM is generally considered to contain four structural domains: The phosphatidylinositol anchor, to which is attached the mannan backbone, followed by the arabinan, and at the non-reducing end of the arabinan various termini commonly known as caps." The authors should consider adding a Scheme here to highlight the basic structure of LAM (including the four structural domains, and the smaller designation of LM within LAM), as well as an indication in these scheme of the specific attachment that is being investigated in this work (i.e., the attachment of the arabinan to the mannan). Most of my comments below, in fact, focus on data aspects because that seems to be the main theme of the paper.

On p. 4, it is stated "Msm-Ara-LM was deacylated and analyzed by high- resolution electrospray liquid chromatography-mass spectrometry (LC-MS) (Fig. 1)"

The authors should add the base peak LC-MS trace to the supplemental information and indicate which four LC peaks correspond to the four types of analytes shown in Figure 1 to allow readers to better understand the caliber of the LC-MS separation of these four mannan backbone variants and to highlight the resolution of separation achieved for these four structurally similar compounds.

For Figure 1 (1) For Figure 1a, the authors should show expansion of M38 non-arabinosylated LM (instead of M37), as this is the most abundant peak in the respective spectrum (as done for subsequent Figures 1b-1d).

(2) The authors should add representative (symbolic) structures for each of the 4 spectra for the most abundant ions to enable readers to better interpret and visualize this data. For example, for Figure 4b, the authors should add the structure of Ara1-LM for the most abundant peak (which is shown in the zoomed region), in this case A1M25. These structures should be provided using the standard symbol nomenclature for glycans (SNFG).

(4) Seeing as this LC-MS data is high-resolution electrospray data and charge states are indicated for all peaks, the authors should add all four deconvoluted spectra to the supplemental information.

(5) The authors should add tables of compound assignments for each assigned peak in Figures 1a-1d. Columns in this table should include experimental masses, theoretical masses, and ppm/Da error assignments to provide confidence in the compound identification for each of these peaks.

On p. 5, this paragraph needs more supporting details and information: "The enzyme digest was then chromatographed on a P-2 column and aliquots of each fractions acetylated and analyzed by LC-MS. The arabinosyl-containing product was found as a trisaccharide of Ara-Man-Man at m/z

912.298 as the NH₄⁺ adduct of the acetylated trisaccharide. Co-eluting with Ara-Man-Man was the disaccharide Man-Man, which evidently was not fully digested to monosaccharides by the αmannosidase. For further purification and structural analysis, the P-4 fractions containing Ara-Man-Man were reduced with NaBD₄, methylated, and identified by LC-MS as the Na⁺ adduct ion at m/z 654.34.

Comments: There is a lot of data described in this paragraph that should be at minimum included in the supplemental information, including:

- (1) Base-peak LC-MS traces of all acetylated fractions
- (2) MS1 spectrum of Ara-Man-Man acetylated trisaccharide NH₄⁺ adduct (including which base-peak LC-MS trace this peak was identified in and at what retention time)
- (3) Base-peak LC-MS traces of all acetylated + then reduced/methylated fractions
- (4) MS1 spectrum of reduced/methylated Ara-Man-Man Na⁺ ion of m/z 654.34 (including which base-peak LC-MS trace this peak was identified in and at what retention time)

All three structural assignments in Figure 2 (parts a,b,c) should have corresponding tables in the supplemental information which include: Ion assignments (C1, B1, etc.), experimental deconvoluted masses, theoretical masses, and ppm error values. The number of decimals in the deconvoluted mass column should reflect the resolution parameters used during MS experiments. Authors should also mention in the main text what limit/threshold for ppm error was used for structural assignments.

Regarding Supplemental Figure 1: "The two hexitols were shown to be in the manno-configuration by comparison with standards."

The last sentence of the supplemental Figure 1 caption is confusing. What kind of comparison was performed between the two mannitol (aka hexitol) samples and standards? Which standards were used? This sounds like an additional experiment which is not explained in detail in this Figure nor mentioned in the main text. Only sample data is shown in this figure, not standard data. Data and figures should be included for any claims of an analysis of configuration via comparison with standards.

Figure 3 should be edited for clarity. It should clearly be indicated that the three spectra shown are simply different m/z ranges of the same spectrum, for example with headers titled "m/z 1100-1300, m/z 1300-1500, m/z 1500-1700". Additionally, a legend should be included for this figure, indicating that blue = A1MX, green=A2MX, pink = MX. A corresponding table should be generated for the supplemental information that supports each peak identification, with columns including: assignment (for example, A1M26), experimental deconvoluted mass, theoretical mass, and ppm error values. Finally, a second supplemental table should be included which shows the base peak LCMS chromatogram for this LC experiment, as well as an indication of the retention time at which the spectrum/spectra shown in Figure 3 were generated.

Page 6

On p. 6, it is stated: "Both, a tri Ara-Man-Man, and tetra AraAra-Man-Man oligosaccharides were identified by LC-MS analysis. LC-MS/MS on the non-derivatized oligosaccharides (Fig. 2b) showed the trisaccharide to be Araf-(1→6)-Manp-(1→6)-Manp and the tetrasaccharide to be Araf-(1→5)-Araf-(1→6)-Manp-(1→6)-Manp."

The organization of the Figures is confusing. Figure 2a is referenced on page 5, then Figure 3 is referenced on page 6, and then Figure 2b is referenced on page 6. Figure 2c is never referenced in the text. Additionally, the authors should include a figure in the supplemental information showing the base peak LC-MS traces for all three MS/MS spectra shown in Figure 2, as well as an indication of the retention times for the three MS/MS spectra.

On p. 7, the authors state: "However, at low mass, low intensity oxonium ions were found as illustrated in Fig. 4. The intensities of low mass oxonium ions in the MS/MS spectrum of the methylated Ara-Man₂₃-Inos-P-Gro triply charged (NH₄⁺)₃ ion at m/z 1784.89 are presented in Table 1."

The scheme shown in Figure 4 does not support this claim. The authors should include in this figure some additional things: the MS1 spectrum showing the triply charged precursor ion of m/z 1784.89 and the MS/MS spectrum of this ion. It is not clear what this scheme is meant to represent. Additionally, Table 1 should be revised to include experimental deconvoluted masses, theoretical masses, and ppm error values for all structural assignments. Additionally, the remainder of this paragraph on page 7 should clearly reference the newly generated figures.

Supplemental Figures 2a and 2b should be cleaned up. Additionally, structures should be shown for each Figure for the two assignments (precursor ion, loss of mannose, loss of arabinosyl residue). Also, the assignments for the mannose cleavage in Figure 2a and the arabinosyl cleavage in Figure 2b are unconvincing, given the low S/N for these two ions. The authors should provide deconvoluted spectra for these ions and provide the S/N value used for deconvolution (preferably S/N = 3 or above).

There is a lot of data missing to support Figure 5. The two base peak LC-MS traces with corresponding MS1 spectra and retention times of these MS1 spectra should be included in the Supplemental Information. The authors should indicate how the area percentages are being calculated – are they being calculated from the base peak LC-MS chromatogram, from XIC LC-MS data, from MS1 spectra, or from deconvoluted MS1 data?

On p. 8, the authors state: "In the case of Ara1Man24-Inos-P-Gro, the MS/MS spectrum of the triple charged $[M+(NH_4)_3]^{3+}$ ion at m/z 1852.9, yielded a doubly charged $(NH_4)_2 Y^-$ ion at m/z 2661.32 (Fig. 6) which corresponds to Ara1Man23-Inos-PGro with a single OH group (see Fig. 2a and Fig. 7) for the formation of the Y ions)."

The ion of m/z 2661.32 should be corrected to 2662.32 (as seen in Figure 6b) such that the main text information matches the m/z labels shown in Figure 6. The same should be done for the remainder of this paragraph (2552.25 \diamond 2553.23, 2559.3 \diamond 2559.26, 2151.1 \diamond 2151.07).

On p. 9, the authors mention: "Two explanations for the quantitatively minor Ara2LM species (see Fig. 1c) are possible.

The authors should include a supplemental Figure to show these two structural possibilities.

For Figure 8, there is a lot of data missing to support this figure. The two base peak LC-MS traces with corresponding MS1 spectra and retention times of these MS1 spectra should be included in the Supplemental Information. Additionally, a supplemental table should be included, with: experimental deconvoluted masses, theoretical masses and ppm error values for all assigned Y-ions. The authors should also add labels to Figure 8 for clarity, for example: orange = M. tuberculosis, blue = M. smegmatis, a) MS/MS spectra of m/z 1906, b) Ara1- Y-ions present in only M. smegmatis, c) Y-ions present in both samples.

On p. 10, the authors indicate: "However, ions were also seen at m/z 's 2559.3, 2457.2, 2355.2 (Fig. 8b)"

The m/z values referenced in the main text should include two numbers after the decimal point for readers to find these peaks in Figure 8, given the complexity of this figure. This comment applies to the entire manuscript.

On p. 15, the authors state: "Typically, analytes (injection volume 1-2 μ l) were injected in 25% solvent B (75% Solvent A) followed by a 30 minutes gradient to 100% solvent B, which was further held for an additional 10 min."

The authors should include the concentrations of all samples analyzed via LC-MS (preferably in ng/ μ L – or the mass of sample per injection: for example, 60 ng) – this should be done for the entire experimental section.

On p. 18, the authors state: "The precursor ion isolation width for MS/MS experiments were set $m/z \pm 4$ Da."

This should be corrected to “[...] were set to ± 4 m/z”. Isolation widths are generally reported in m/z units, not in mass units (Da).

Reviewer #1:

In this work, Angala and co-workers, determine that the arabinan component of the mycobacterial lipoarabinomannan (LAM) is attached at the nonreducing end of the mannan rather than to internal regions. In addition, the authors identified the presence of secondary extended mannan side chains attached to the internal mannan region, leading not only to a significant revision of the chemical structure of LAM, but also to the structural and functional role of the lypoglycan both in the periplasm and outside the cell envelope. The manuscript is clear and well written. Because of its novelty and the quality of the work, I support the publication of this manuscript in Communications Chemistry.

We thank the reviewer for his/her positive feedback.

Reviewer #2:

The manuscript entitled “The Presence of Secondary Extended Mannan Side Chains and the Attachment of the Arabinan in Mycobacterial Lipoarabinomannan” by Angala et al. it is an expansion of a previous report describing how the arabinan domain of the lipoarabinomannan molecule attaches to the mannan domain. In addition, it also describes the novel finding of the existence of a secondary extended branched pentasaccharide mannose branch from the mannan-core. This an interesting report revisiting the structure of LAM and LM from a very well-established group in the biosynthesis of the mycobacterial cell wall. There are some suggestions that will excel the quality of this reported study:

It is not clear how many times these experiments were performed and how data are being reported (means?). And if it is a mean value, where are the SD or SEM? This is obvious in Fig 5, where it is not clear how many times this has been done and if the digestion was exhaustive or not.

The α -mannosidase hydrolysis of *M. tuberculosis* LAM was from two independent experiments and the results from one representative experiment are shown. Digestion of Msm-Ara-LM from *M. smegmatis* Δ MSMEG-6387 was done once. This information has been added to the Figure 5 (now Fig. 6) legend. In the case of Msm-Ara-LM, digestion with α -mannosidase (Jack bean) was exhaustive. After overnight digestion, we replenished with fresh α -mannosidase (15 units/mg protein) and further digested for an additional 8 hr as described in the Material and Methods. We monitored the digestion by acquiring LC-MS spectra at different time points and noticed no further changes in digestion pattern after overnight treatment. The same conditions were applied to the *M. tuberculosis* material.

Some of the m/z values mentioned in the text do not correspond to the exact values in the figure, please revise (page 8 reports m/z 2661.32 but in figure is m/z 2262.32), the gain/loss of a proton needs to be accounted. If this is not the case, clarify why.

For consistency, as pointed out by reviewers # 2 and 3, we corrected the m/z values corresponding to ^{12}C throughout the text.

Revise Fig 7c, if I am not mistaken, the m/z value depicted for Y21 (-OH)₁ should be m/z 2458.21 in place of m/z 2559.

Thank you so much for pointing this error. We corrected this in Figure 7.

It will be interesting to determine or comment if specific acylations (e.g. succinates) indicate stooping points or branching for mannosyl- and arabinosyl-transferases.

The physiological significance of succinates is currently not well understood. The fact that succinyl residues modify the C2 position of a portion of the internal α -3,5-branched Araf residues of LAM and arabinogalactan (AG) could indeed suggest that they act as molecular signals for the branching of the arabinan domains of these two polysaccharides. However, our recent work on a succinylation-deficient mutant of *Mycobacterium smegmatis* argues against this hypothesis in that the LAM and AG produced by the mutant presented no significant structural alterations (Palcekova *et al.*, 2019, *J. Biol. Chem.*, Vol. 294; pp. 10325-10335). Moreover, succinates also modify quantitatively minor α -1,5-Araf positions of the arabinan domains of AG and LAM from *M. tuberculosis* and *M. smegmatis*, in addition to modifying the C3 position of β -(1 \rightarrow 2)-linked Araf residues of the non-reducing arabinan termini of LAM in *M. tuberculosis*. In light of the recent observation that the prevalence of succinyl residues on *M. tuberculosis* LAM increase during host infection (De *et al.*, 2020; *ACS Infect. Dis.*, Vol. 6, pp. 291-301), it is tempting to speculate that the biological significance of this motif essentially resides in the modulation of host-pathogen interactions, a hypothesis that our laboratory is currently testing.

In the discussion, the authors need to expand about the biological meaning of their results, what these may mean knowing how *M. tuberculosis* uses this molecule as a virulent factor the gain entry and survive in host cells.

We have now expanded in the discussion of the possible biological meaning of our results (p. 14, one but last paragraph).

Does the LAM heterogeneity only has to do with different stages of LAM production at the time that LAM is being purified or is the LAM structure dynamic?

Purified LAM, independent of growth phase, always comes out as a heterogeneous population of lipoglycans differing in their degree of acylation, glycosylation (including capping motifs) and charge (succinylation). This being said, the amount of LAM produced by mycobacteria, its composition and subcellular localization are also known to vary with growth phase (e.g., Yang *et al.*, 2013 - Changes in the major cell envelope components of *Mycobacterium tuberculosis* during *in vitro* growth. *Glycobiology* **23**, 926-934).

The last sentence in the discussion: To which purpose?

We have modified the end of the discussion.

The blue labeling in Fig 1 as depicted may be confusing to the general reader, as increments do not seem to correspond to a hexose. Clarify in the figure legend.

We edited this figure as suggested by the reviewer.

Fig 2c needs to be referenced in the text.

Thank you so much for pointing this error; Fig. 2c is now referenced in the text on page 6.

Fig 9 specify from which mycobacterial species this LAM structure is from, not PI-caps or Man-caps are not depicted.

ManLAM from *M. tuberculosis* H37Rv is represented in this Figure as now clearly stated in the legend.

Consider to change: This data suggests... to These data suggest...

Changed.

Reviewer #3:

This paper uses high resolution mass spectrometry to characterize lipoarabinomannan, an arabinosylated lipopolysaccharide of mycobacteria. The authors identify previously undiscovered mannan side chains, and utilize various enzymatic procedures to help map the composition of the lipoarabinomannan. The results have the potential to be exciting to a certain part of the bio community; however, the current state of the manuscript falls short in the quality of the figures, the lack of rigor in the supporting information, and in some cases the description of the data and interpretation. Moreover, although the depth of analysis is impressive, the impact of the paper is overshadowed by the focus on extreme data analysis coupled with lack of appropriate supporting data.

On page 2, the authors indicate “LAM is generally considered to contain four structural domains: The phosphatidylinositol anchor, to which is attached the mannan backbone, followed by the arabinan, and at the non-reducing end of the arabinan various termini commonly known as caps.”

The authors should consider adding a Scheme here to highlight the basic structure of LAM (including the four structural domains, and the smaller designation of LM within LAM), as well as an indication in these scheme of the specific attachment that is being investigated in this work (i.e., the attachment of the arabinan to the mannan). Most of my comments below, in fact, focus on data aspects because that seems to be the main theme of the paper.

As per the reviewer’s suggestion, we included a new Figure (Figure 1) representing the four basic structural domains of lipoarabinomannan using the symbol nomenclature for glycans.

On p. 4, it is stated “Msm-Ara-LM was deacylated and analyzed by high- resolution electrospray liquid chromatography-mass spectrometry (LC-MS) (Fig. 1)”

The authors should add the base peak LC-MS trace to the supplemental information and indicate which four LC peaks correspond to the four types of analytes shown in Figure 1 to allow readers to better understand the caliber of the LC-MS separation of these four mannan backbone variants and to highlight the resolution of separation achieved for these four structurally similar compounds.

As per the reviewer’s suggestion, we included a supplementary Figure 1, showing BPC along with EIC corresponding to the dominant species shown in each Fig. 1 panel (a-d) (now Fig. 2a-d).

For Figure 1 (1) For Figure 1a, the authors should show expansion of M38 non-arabinosylated LM

(instead of M37), as this is the most abundant peak in the respective spectrum (as done for subsequent Figures 1b-1d).

We agree with the reviewer, and for consistency in Fig 1 (a) (now Fig. 2(a)), we added the expansion of M38 and removed M37.

(2) The authors should add representative (symbolic) structures for each of the 4 spectra for the most abundant ions to enable readers to better interpret and visualize this data. For example, for Figure 4b, the authors should add the structure of Ara1-LM for the most abundant peak (which is shown in the zoomed region), in this case A1M25. These structures should be provided using the standard symbol nomenclature for glycans (SNFG).

“For example, for Figure 4b”: we assume the reviewer is talking about Figure 1b (not 4b).

As per the reviewer’s suggestion, we included schematic structures for all four species in Fig. 1 (now Fig. 2).

(4) Seeing as this LC-MS data is high-resolution electrospray data and charge states are indicated for all peaks, the authors should add all four deconvoluted spectra to the supplemental information.

All four deconvoluted spectra are now shown in Supplementary Figure 1(f-i).

(5) The authors should add tables of compound assignments for each assigned peak in Figures 1a-1d. Columns in this table should include experimental masses, theoretical masses, and ppm/Da error assignments to provide confidence in the compound identification for each of these peaks.

This information is now presented in Supplementary Table 1.

On p. 5, this paragraph needs more supporting details and information: “The enzyme digest was then chromatographed on a P-2 column and aliquots of each fractions acetylated and analyzed by LC-MS. The arabinosyl-containing product was found as a trisaccharide of Ara-Man-Man at m/z 912.298 as the NH₄⁺ adduct of the acetylated trisaccharide. Co-eluting with Ara-Man-Man was the disaccharide Man-Man, which evidently was not fully digested to monosaccharides by the αmannosidase. For further purification and structural analysis, the P-4 fractions containing Ara-Man-Man were reduced with NaBD₄, methylated, and identified by LC-MS as the Na⁺ adduct ion at m/z 654.34.

Comments: There is a lot of data described in this paragraph that should be at minimum included in the supplemental information, including:

- (1) Base-peak LC-MS traces of all acetylated fractions
- (2) MS1 spectrum of Ara-Man-Man acetylated trisaccharide NH₄⁺ adduct (including which base-peak LC-MS trace this peak was identified in and at what retention time)
- (3) Base-peak LC-MS traces of all acetylated + then reduced/methylated fractions
- (4) MS1 spectrum of reduced/methylated Ara-Man-Man Na⁺ ion of m/z 654.34 (including which base-peak LC-MS trace this peak was identified in and at what retention time)

- (1) Base-peak LC-MS traces of all acetylated fractions

We provided this information in Supplementary Figure 2(a), but we have not shown LC/MS data for all the fractions as it does not add much information; instead, we chose to present every alternate fraction.

(2) MS1 spectrum of Ara-Man-Man acetylated trisaccharide NH₄⁺ adduct (including which base-peak LC-MS trace this peak was identified in and at what retention time)

Included in the Supplementary Figure 2 (b).

(3) Base-peak LC-MS traces of all acetylated + then reduced/methylated fractions

(4) MS1 spectrum of reduced/methylated Ara-Man-Man Na⁺ ion of m/z 654.34 (including which base-peak LC-MS trace this peak was identified in and at what retention time)

Showing the BPC traces for all the reduced/methylated fractions would be redundant with Supplementary Figure 2 (a). Instead, we chose to provide the TIC, EIC, and MS1 spectrum along with RT in Supplementary Figure 3 (a-b).

All three structural assignments in Figure 2 (parts a,b,c) should have corresponding tables in the supplemental information which include: ion assignments (C1, B1, etc.), experimental deconvoluted masses, theoretical masses, and ppm error values. The number of decimals in the deconvoluted mass column should reflect the resolution parameters used during MS experiments. Authors should also mention in the main text what limit/threshold for ppm error was used for structural assignments. – We provided the above information in the Supplementary Table 2.

Regarding Supplemental Figure 1: “The two hexitols were shown to be in the manno-configuration by comparison with standards.”

The last sentence of the supplemental Figure 1 caption is confusing. What kind of comparison was performed between the two mannitol (aka hexitol) samples and standards? Which standards were used? This sounds like an additional experiment which is not explained in detail in this Figure nor mentioned in the main text. Only sample data is shown in this figure, not standard data. Data and figures should be included for any claims of an analysis of configuration via comparison with standards.

Additional TICs and EICs corresponding to the standard 1,6 linked mannobiose were included for comparison with the sample data in Supplementary Figure 4(b).

On p. 7, the authors state: “However, at low mass, low intensity oxonium ions were found as illustrated in Fig. 4. The intensities of low mass oxonium ions in the MS/MS spectrum of the methylated Ara-Man₂₃-Inos-P-Gro triply charged (NH₄⁺)₃ ion at m/z 1784.89 are presented in Table 1.” The scheme shown in Figure 4 does not support this claim. The authors should include in this figure some additional things: the MS1 spectrum showing the triply charged precursor ion of m/z 1784.89 and the MS/MS spectrum of this ion. It is not clear what this scheme is meant to represent. Additionally, Table 1 should be revised to include experimental deconvoluted masses, theoretical masses, and ppm error values for all structural assignments. Additionally, the remainder of this paragraph on page 7 should clearly reference the newly generated figures.

“It is not clear what this scheme is meant to represent”: The scheme represents the understanding of mannosyl residues substitution next to the site of arabinose attachment at the non-reducing end of the mannan backbone.

We believe the modified Fig. 4, (now Fig. 5) with corresponding MS/MS spectrum, and schematic structures in Supplementary Figure 6 now makes it clearer. Table 1 has been updated with mass accuracy and mass error values, as requested by the reviewer.

Supplemental Figures 2a and 2b should be cleaned up. Additionally, structures should be shown for each Figure for the two assignments (precursor ion, loss of mannose, loss of arabinosyl residue). Also, the assignments for the mannose cleavage in Figure 2a and the arabinosyl cleavage in Figure 2b are unconvincing, given the low S/N for these two ions. The authors should provide deconvoluted spectra for these ions and provide the S/N value used for deconvolution (preferably S/N = 3 or above).

As per the reviewer’s suggestion, we added schematic structures to Supplementary Figures 2a and 2b, (now Supplementary Figure 7a and 7b). The revised text makes Figures 7a and 7b less confusing, indicating the lack of ions corresponding to the cleavage of mannosyl and arabinosyl residues.

There is a lot of data missing to support Figure 5. The two base peak LC-MS traces with corresponding MS1 spectra and retention times of these MS1 spectra should be included in the Supplemental Information.

Figure 5 is now Figure 6. We provided the missing data in Supplementary Figure 8.

The authors should indicate how the area percentages are being calculated – are they being calculated from the base peak LC-MS chromatogram, from XIC LC-MS data, from MS1 spectra, or from deconvoluted MS1 data?

We used intensities from the MS1 spectrum. The area percentages are included in the attached excel sheet as Supplementary Data 2.

On p. 8, the authors state: “In the case of Ara1Man24-Inos-P-Gro, the MS/MS spectrum of the triple charged $[M+(NH_4)_3]^{3+}$ ion at m/z 1852.9, yielded a doubly charged $(NH_4)_2^{2+}$ Y- ion at m/z 2661.32 (Fig. 6) which corresponds to Ara1Man23-Inos-PGro with a single OH group (see Fig. 2a and Fig. 7) for the formation of the Y ions.” The ion of m/z 2661.32 should be corrected to 2662.32 (as seen in Figure 6b) such that the main text information matches the m/z labels shown in Figure 6. The same should be done for the remainder of this paragraph (2552.25 \diamond 2553.23, 2559.3 \diamond 2559.26, 2151.1 \diamond 2151.07).

We edited this as suggested by the reviewer.

On p. 9, the authors mention: “Two explanations for the quantitatively minor Ara2LM species (see Fig. 1c) are possible. The authors should include a supplemental Figure to show these two structural possibilities.

No additional figures were generated but we provided the reference figures (Fig. 8e and Fig. 3c) in the text on page 9.

For Figure 8, there is a lot of data missing to support this figure. The two base peak LC-MS traces with corresponding MS1 spectra and retention times of these MS1 spectra should be included in the Supplemental Information.

Provided in the Supplementary Figure 9.

Additionally, a supplemental table should be included, with: experimental deconvoluted masses, theoretical masses and ppm error values for all assigned Y-ions.

Provided in Supplementary Table 4.

The authors should also add labels to Figure 8 for clarity, for example: orange = *M. tuberculosis*, blue = *M. smegmatis*,

We edited Figure 8 (now Figure 9) as suggested by the reviewer.

a) MS/MS spectra of m/z 1906, b) Ara1- Y-ions present in only *M. smegmatis*, c) Y-ions present in both samples.

This is now clearly described in the figure legend.

On p. 10, the authors indicate: “However, ions were also seen at m/z’s 2559.3, 2457.2, 2355.2 (Fig. 8b)”

The m/z values referenced in the main text should include two numbers after the decimal point for readers to find these peaks in Figure 8, given the complexity of this figure. This comment applies to the entire manuscript.

We edited the entire manuscript as suggested by the reviewer.

On p. 15, the authors state: “Typically, analytes (injection volume 1-2 μ l) were injected in 25% solvent B (75% Solvent A) followed by a 30 minutes gradient to 100% solvent B, which was further held for an additional 10 min.” The authors should include the concentrations of all samples analyzed via LC-MS (preferably in ng/ μ L – or the mass of sample per injection: for example, 60 ng) – this should be done for the entire experimental section.

We provided concentrations in ng/ μ l

On p. 18, the authors state: “The precursor ion isolation width for MS/MS experiments were set m/z \pm 4 Da.”

This should be corrected to “[...] were set to \pm 4 m/z”. Isolation widths are generally reported in m/z units, not in mass units (Da).

We corrected this as suggested by the reviewer.

REVIEWERS' COMMENTS:

Reviewer #2 (Remarks to the Author):

The authors properly addressed all my previous comments/concerns. This is a well-performed and comprehensive study from a world-recognized group working for many years on the mycobacterial cell wall structure.

Reviewer #3 (Remarks to the Author):

The revised manuscript is in good shape and represents a high level, proficiently executed structural characterization of mycobacterial lipoarabinomannan. The new supplemental materials and more explicit annotation are very helpful. I recommend acceptance without further review. The authors are commended for their diligence with the extensive reviews.